# Effect of safranal or candesartan on 3-nitropropionicacid-induced biochemical, behavioral and histological alterations in a rat model of Huntington's disease

**Nagwa Ibrahim Shehata**[1], **Dina Mohamed Abd EL-Salam**[2], **Roqaya Mahmoud Hussein**[2], **Sherine Maher Rizk**[1] *

**1** Faculty of Pharmacy, Biochemistry Department, Cairo University, Cairo, Egypt, **2** Hazards Department, Egyptian Drug Authority (EDA), Cairo, Egypt

* sherine.abdelaziz@cu.edu.eg

**Data Availability Statement:** The data available on the repository of figshare under DOI number of: 10.6084/m9.figshare.23650620.

## Abstract

3-nitropropionic acid (3-NP) is a potent mitochondrial inhibitor mycotoxin. Systemic administration of 3-NP can induce Huntington's disease (HD)-like symptoms in experimental animals. Safranal (Safr) that is found in saffron essential oil has antioxidant, anti-inflammatory and anti-apoptotic actions. Candesartan (Cands) is an angiotensin receptor blocker that has the potential to prevent cognitive deficits. The present study aims to investigate the potential neuroprotective efficacy of Safr or Cands in 3-NP-induced rat model of HD. The experiments continued for nine consecutive days. Rats were randomly assigned into seven groups. The first group (Safr-control) was daily intraperitoneally injected with paraffin oil. The second group (Cands- and 3-NP-control) daily received an oral dose of 0.5% carboxymethylcellulose followed by an intraperitoneal injection of 0.9% saline. The third and fourth groups received a single daily dose of 50 mg/kg Safr (intraperitoneal) and 1 mg/kg Cands (oral), respectively. The sixth group was daily treated with 50 mg Safr kg/day (intraperitoneal) and was intraperitoneally injected with 20 mg 3-NP/ kg, from the 3$^{rd}$ till the 9$^{th}$ day. The seventh group was daily treated with 1 mg Cands /kg/day (oral) and was intraperitoneally injected with 20 mg 3-NP/ kg, from the 3$^{rd}$ till the 9$^{th}$ day. The present results revealed that 3-NP injection induced a considerable body weight loss, impaired memory and locomotor activity, reduced striatal monoamine levels. Furthermore, 3-NP administration remarkably increased striatal malondialdehyde and nitric oxide levels, whereas markedly decreased the total antioxidant capacity. Moreover, 3-NP significantly upregulated the activities of inducible nitric oxide synthase and caspase-3 as well as the Fas ligand, in striatum. On the contrary, Safr and Cands remarkably alleviated the above-mentioned 3-NP-induced alterations. In conclusion, Safr and Cands may prevent or delay the progression of HD and its associated impairments through their antioxidant, anti-inflammatory, anti-apoptotic and neuromodulator effects.

**Funding:** The author(s) received no specific funding for this work.

**Competing interests:** The authors have declared that no competing interests exist.

## Introduction

Huntington's disease (HD) is a fatal neurodegenerative disease characterized by significant motor and cognitive impairments. It is associated with psychiatric and behavioral disturbances. Unfortunately, no cure has been recognized yet to slow down or reverse its progression [1,2]. Pathologically, HD is caused by an expansion of cytosine–adenine–guanine (CAG) trinucleotide repeats in huntingtin gene exon causing an abnormal translation of its protein [3,4]. The mutant huntingtin protein causes disruption of the striatum initially followed by the cortex and the hippocampus [5]. Several mechanisms were suggested to explain the pathogenesis of HD such as oxidative stress, mitochondrial or synaptic dysfunction, defects in energy metabolism and apoptosis of striatal neurons, but the exact mechanism is not clear so far [6].

3-Nitropropionic acid (3-NP) is a natural toxin obtained from various plants and fungi [7]. It is a potent mitochondrial inhibitor mycotoxin. Systemic administration of 3-NP to experimental animals produces HD-like symptoms via developing bilateral striatal-specific lesions. 3-NP is a non-competitive inhibitor of mitochondrial succinate dehydrogenase (SDH, complex II), interfering with the electron transport cascade and oxidative phosphorylation, resulting in cellular energy deficit and oxidative stress [8].

The use of medicinal plants rich in antioxidant phytochemicals as potential protective agents against several brain diseases has received growing attention. Their protective mechanism is via scavenging reactive oxygen species (ROS) and detoxifying potent genotoxic oxidants. Saffron is the dry red stigma of *Crocus sativus L.*, Iridaceae, one of the most expensive herbs used as a natural additive in cooking to enhance flavor, color and aroma. It is used also in the traditional medicine for its hypolipidemic, anti-cancer and antidepressant properties [9]. Safranal (Safr) is the main component of saffron essential oil that causes its distinctive aroma. It has been shown that Safr has several pharmacological effects such as antioxidant, anti-inflammatory, gastro-protective and anti-apoptotic effects [10]. Furthermore, Safr could improve cognitive and memory deficits and exerts numerous neuropharmacological properties as anxiolytic, anticonvulsant and antidepressant [11]. Several studies have reported that Safr may be an effective treatment for various neurodegenerative diseases such as Parkinson's disease [12] and Alzheimer's disease [13].

The renin-angiotensin system (RAS) is one of the most important systems in the body that regulates blood pressure and fluid homeostasis [14]. Angiotensin II, the main effector in RAS, contributes mainly to the impairment of neurovascular coupling impairments [15]. Ang II type 1 receptor (AT1R) blockers (ARBs) have been reported to be useful in diminishing cognitive deficits linked to Post-Stroke Cognitive Impairment (PSCI), Alzheimer's Disease, Parkinson's Disease, and Vascular Cognitive Impairment (VCI) [16]. A recent study reported that dysregulated brain RAS may be implicated in neurodegeneration due to neuroinflammation, oxidative stress and aging-related pathophysiological changes [17].

ARBs can also ameliorate irradiation-induced brain stroke via protecting the cerebrovascular flow and reducing the risk of cognitive impairment [18]. Candesartan (Cands), the strongest AT1R antagonist can cross the blood-brain barrier to prevent traumatic brain injury and ischemia [19]. Cands represents a promising possibility for the treatment and prevention of age-related memory impairment [20].

In the light of the above-mentioned beneficial effects, we hypothesize that the use of Safr or Cands may possibly ameliorate 3-NP-induced striatal damage in rats. To our knowledge, no study was designed to study the potential neuroprotective effect of Cands against 3-NP induced HD symptoms. However, only one recent study aimed to only investigate the effect of Safr on the 3-NP induced behavioral and oxidative stress parameters in HD rat model [21]. Thus, the present study aimed at investigating the possible protective effect of Safr and Cands

against 3-NP induced mitochondrial, biochemical, behavioral and histological changes in the striatum of male Wistar rats.

## Materials and methods

### Chemicals

Pure Safranal (Safr), Cands (Candesartan), 3-NP (3-nitropropionic acid), paraffin oil, 0.5% carboxy methyl cellulose (CMC), thiobarbituric acid (TBA), vanadium chloride, sulfanilamide, N-(1-naphthyl) ethylenediamine, Folin-Ciocalteau reagent and bovine serum albumin (BSA) were obtained from Sigma (St. Louis, MO, USA). All other chemicals were of the highest analytical grades commercially available.

### Dose preparation of drugs

**Preparation of 3-NP.** To prepare a dose of 3-NP (20 mg/kg/day), each 20 mg of 3-NP was dissolved in 2.0 mL of 0.9% saline.

**Preparation of Safr.** A 50 mg of Safr was dissolved in 2.0 mL of paraffin oil to prepare the dose of Safr 50 mg/kg/day.

**Preparation of Cands.** Finally, 1 mg of Cands was dissolved in 2.0 mL of 0.5% CMC to prepare the dose of Cands 1 mg/kg/day.

For all the applied drugs, the volume of administration was adjusted to 2.0 mL/kg body weight.

**Animals and experimental procedure.** Adult male Wistar rats, 7–8 weeks old, weighing $250 \pm 25$g were obtained from the animal house of the National Organization for Drug Control and Research, Egypt. During acclimatization period, the rats were housed in Polypropylene cages (37x 24 18 $cm^3$), with 4 rats per cage. The rats were kept under controlled conditions of constant temperature ($23 \pm 2°C$), humidity ($55 \pm 5\%$) and 12 h/12h light/dark cycle (light on from 6:00 am to 5:59 pm and light off from 6:00 pm to 5:59 am), with *ad libitum* access to standard rodent chow and water. After one week of acclimatization, 280 rats were divided into seven main groups with 40 rats per group, as follows (Fig 1):

- The first group was intraperitoneally (IP) injected with paraffin oil (PO), daily for 9 days, as negative control group for Safr group.

- The second group was pretreated orally with 0.5% CMC, daily for 9 days and was intraperitoneally injected with saline daily from the 3rd day to the 9th day, as negative control group for Cands and 3-NP groups.

- The third group (Safr-treated) received intraperitoneal injection of Safr 50 mg/kg/day for 9 days [22].

- The fourth group (Cands-treated) was orally administered with Cands 1 mg/kg/day for 9 days [23].

- The fifth group (3-NP) received IP injection of 3-NP 20 mg/kg/day from the 3rd till the 9th day [24].

- The rats of the sixth group (Safr+3-NP) were intraperitoneally injected with Safr 50 mg/kg/day from the 1st till 9th day, whereas intraperitoneally injected with 3-NP 20 mg/kg/day from the 3rd day to the 9th day.

- The rats of the seventh group (Cands+3-NP) were orally administered with Cands 1 mg/kg/day from the 1st till 9th day whereas intraperitoneally injected with 3-NP 20 mg/kg/day from the 3rd day to the 9th day.

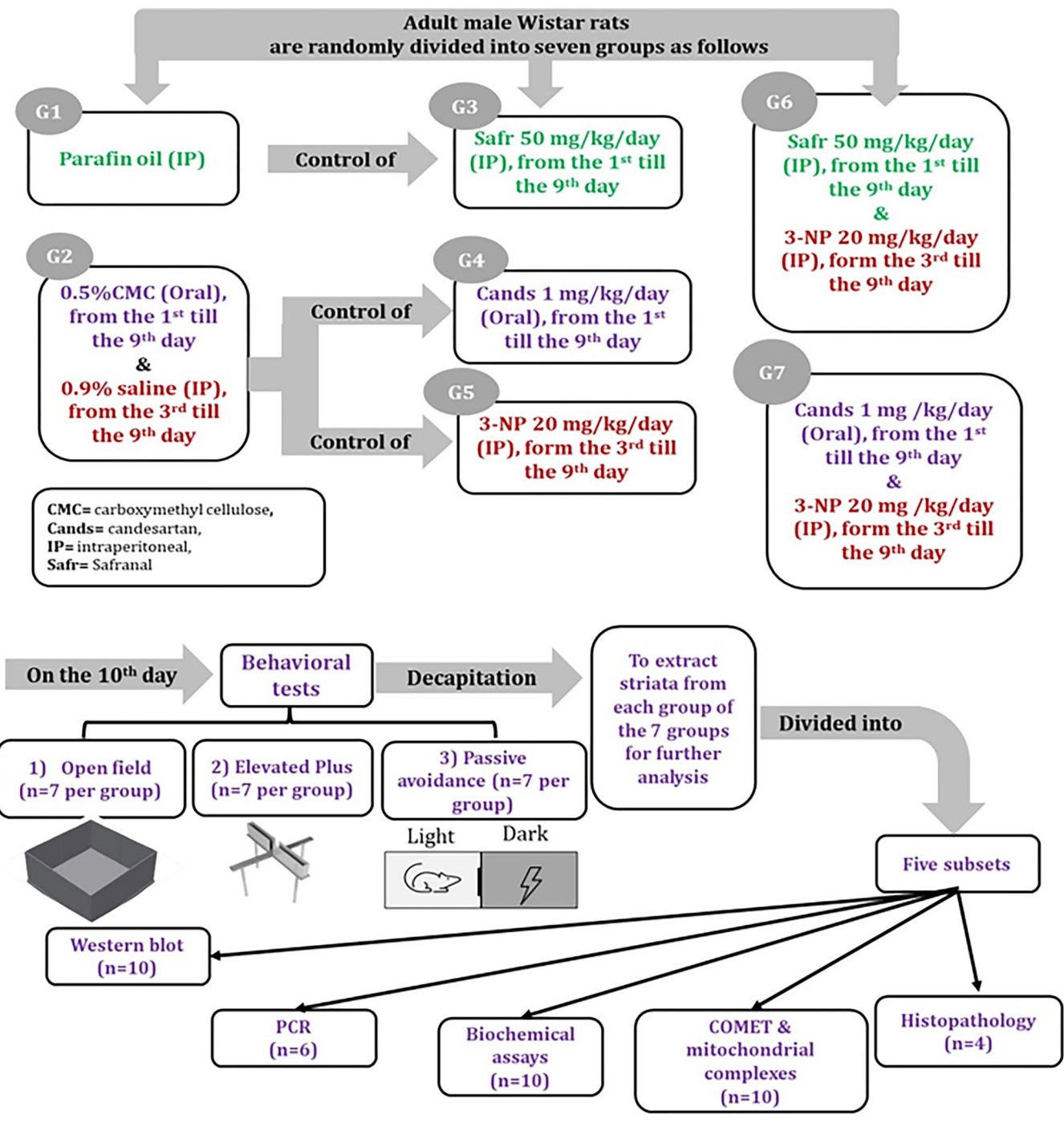

**Fig 1. Summary of the experimental design.**

At the end of the experimental period, behavioral tests (elevated plus maze "EPM", open field task "OFT" and passive avoidance "PA") were executed. In addition, the total body weight was recorded. Rats were sacrificed by decapitation and the striata were separated from the isolated brains. Tissue samples were stored at −80°C for further analysis.

The current investigation complies with the Guide for the Care and Use of Laboratory Animals published by the US National Institute of Health (NIH Publication No. 85–23, revised 2011) and was approved by the Ethical Committee for Animal Experimentation at the Faculty of Pharmacy, Cairo University, approval no.: BC (1235).

## Behavioral tests

**Open field test (OFT):** The test was carried out to assess possible effects on locomotor activity during the morning daylight in a quiet lab, to avoid interference with any external stimuli. The OFT was executed in a square wooden arena ($100 \times 100 \times 25$ cm$^3$). The clean floor was divided by black lines into 25 small squares ($20 \times 20$ cm$^2$) [8]. The test started on the day 10 after treatment, at 8.00 am. Rats were placed individually into the central point of the open field and observed during five minutes test session for latency period "the time interval in seconds (s) between placing the animal at the middle of the arena until the decision of the animal to move", ambulation frequency "the number of squares crossed by the animal per minute", grooming frequency "the number of face washing and scratching with the hind leg and licking of the fur and genitals per 5 min" and rearing frequency "the number of times the animal stood stretched on the hind limbs with or without fore limbs support per 5 min". After each session, the floor of the OF was cleaned accurately from the urine spots and fees with 70% alcohol.

**Elevated plus maze test:** The test was used to examine anxiety and spontaneous motor activity. The stainless-steel apparatus consists of 4 crossed arms that painted with matte black. Two open arms ($50 \times 10 \times 30$ cm$^3$) and two closed arms ($50 \times 10 \times 30$ cm$^3$) and the maze was elevated 65 cm above the floor. The test started on the day 10, after treatment, at 1:00 pm. The rat was placed in the center of the maze and the number of entries in open and closed arms, as well as the time the animal spent in the open and closed arms during a period of 5 minutes test session were recorded. The degree of avoidance of the open arms of the maze has been considered as a measure of strength of fear drive [20,25].

**Passive avoidance test:** to evaluate learning and spatial memory, an apparatus consisted of two equal-sized compartments ($25 \times 25 \times 25$ cm$^3$), including a light and a dark compartment with a grid floor and Plexiglas walls that were separated by a guillotine door was used. To accustom rats to the apparatus, all rats were placed into the shuttle box for 1 day before main experiment for five min. On the 9$^{th}$ day of treatment, training session started at 6:00 pm. In the training session, each rat was individually put into the light compartment for 60s. After opening the guillotine door and the entrance of the rat into the dark compartment, the door was closed and a 0.5 mA with 50 Hz foot electric shock was delivered for 2s, through the grid floor. After 20s, the rats were transferred to their cage. On the 10$^{th}$ day of treatment, at 6:00 pm, the test session was done. In the test session, rats were placed in the light compartment. The step-through latency to enter the dark compartment was measured as a positive index of memory function. The delay in entering the dark compartment was recorded to a maximum of 3 minutes. The latency to enter the darkened chamber was measured for both acquisition (time from first exposure to the lighted chamber until the rat entered the darkened chamber) and retention (time from entry to lighted chamber until the rat entered the darkened chamber 24 hours later) [26].

## Tissue preparation and sample analysis

Each experimental group was divided into 5 subsets; the first subset (n = 10) was used to assess striatal caspase-3 (Casp-3), Fas ligand (Fas-L) and inducible nitric oxide synthase (iNOS) protein levels by Western blot technique. The second subset (n = 6) was used to measure the gene expression level of monoamine oxidases (MAO-A) and (MAO-B) via quantitative reverse

transcription polymerase chain reaction (qRT-PCR) technique. The third subset (n = 10) was used for estimating oxidative stress parameters, monoamine levels and activities of MAO and acetylcholinesterase (AChE). The fourth subset (n = 10) was used for the analysis of mitochondrial complexes and comet assay. The last subset (n = 4) was used for striatal histopathological examination.

## Biochemical analyses

Striatum was weighed and 10% (w/v) homogenate was prepared in ice-cold 1.15% potassium chloride (KCl) for the determination of oxidative stress parameters. Malondialdehyde (MDA) content was determined as a biomarker of lipid peroxidation, using thiobarbituric acid (TBA) as described by Uchiyama and Mihara [27]. Nitric oxide (NO) level was determined by Griess method [28]. Total antioxidant capacity (TAC) was measured using a commercial colorimetric kit (Biodiagnostic, Egypt) according to the manufacturer's instructions.

In addition, striatal tissue was homogenized in cold acidified n-butanol for analyzing AChE activity [29], MAO activity [30] and monoamine levels (dopamine, DA, norepinephrine, NE, serotonin "5-hydroxytryptamine; 5-HT" and 5-hydroxyindoleacetic acid, 5-HIAA) [31], using the fluorometric methods.

## QRT-PCR analysis of MAO-A and MAO-B

Total ribonucleic acid (RNA) was isolated from brain striatum using Qiagen tissue extraction kit (USA) according to the manufacturer's instructions. The concentration and quality of the obtained RNA were assessed spectrophotometrically through measurement of the A260/A280 ratio using NanoDrop2000 (Thermo, USA). The total RNA was used for complementary DNA (cDNA) conversion using high-capacity cDNA reverse transcription kit (Fermentas, USA) according to the manufacturer's instructions. Quantitative real time PCR (qPCR) was performed using SYBR Green PCR Master Mix (Applied Biosystems, USA) with software version 3.1 (StepOne™, USA). The qPCR assay with the primer sets were optimized at the annealing temperature. The sequences of the primers were as follows:

| Gene | Primer sequence |
|---|---|
| MAO-A | F: 5′-TGACCCAGTATGGAAGGGTGAT-3′ |
| | R: 5′-TCTGTGCCTGCAAAGTAAATCC-3′ |
| MAO-B | F: 5′-ATGAGCAACAAAAGCGATGTGA-3′ |
| | R: 5′- TCCTAATTGTGTAAGTCCTGCCT-3′ |
| β-actin (housekeeping gene). | F: 5′-GGCTGTATTCCCCTCCATCG-3′ |
| | R: 5′-CCAGTTGGTAACAATGCCATGT-3′ |

The thermal cycle protocol consisted of initial denaturation at 95˚C for 10 min followed by 40 cycles with 15s denaturation at 95˚C and 1 min annealing at 60˚C /extension at 72˚C. The changes in target gene expression were calculated using the comparative threshold cycle (Ct) method and presented as fold change [32].

## Estimation of Casp-3, Fas-L and iNOS proteins by Western blot technique

The western blot method was done using V3 Western Workflow™ Complete System, Bio-Rad℞ Hercules, California, USA). Briefly, 5 mg of striatum tissue was homogenized in Radio-immunoprecipitation assay (RIPA) buffer, then centrifuged at 12,000 rpm for 20 min. The protein concentration for each cell lysate was determined using Bradford assay. Equal amounts

of protein (20–30 μg of total protein from cell lysate) were separated by sodium dodecyl-sulfate polyacrylamide gel electrophoresis (SDS-PAGE) and then transferred to a polyvinylidene difluoride membrane. The membrane was blocked in Tris-buffered saline (TBS) buffer, 3% Bovine serum albumin (BSA) and 0.1% Tween 20 at room temperature for 1 h and incubated with Casp-3, Fas-L and iNOS primary antibodies, supplied by Thermoscientific (Loughborough, UK), overnight at pH 7.6 at 4°C with gentle shaking. After washing, peroxidase-labeled secondary antibodies were added, and the membranes were incubated at room temperature for 1h. Image analysis software was used to read the band intensity of the target proteins against control sample after normalization by β-actin on the Chemi Doc MP imager.

## Measurement of mitochondrial complexes I, II and IV activities

**Complex I activity.** Complex I (NADH:ubiquinone oxidoreductase) activity was measured using mitochondrial Complex I Activity Colorimetric Assay Kit (Bio-vision, Milpitas, California, USA), according to Ansari et al. [33]**.** The kit uses decylubiquinone, an analog of ubiquinone, as an electron acceptor that gets converted to decylubiquinol through the catalytic activity of complex I. The complex I dye absorbs light at 600 nm in its oxidized form and accepts electrons from decylubiquinol. Complex I activity was determined colorimetrically by recording the change in absorbance of reduced complex I dye at 600 nm. Complex I activity was calculated from the equation:

$$\text{Complex I activity (mUnits/}\mu g\text{)} \frac{\Delta \,[\text{reduced complex I dye concentration}]}{\Delta t \; x \; p \; x \; D}$$

where Δ [reduced complex I dye concentration] is the change in reduced complex I dye concentration during Δt, Δt = $t_2 - t_1$ (min), p is the mitochondrial protein (μg) and D is the dilution factor.

Then the net complex I activity in the sample was calculated by subtracting the activity in reaction without rotenone minus the activity in reaction with rotenone. One unit of complex I is the amount of enzyme that causes the reduction of 1 μmol of the dye per min at pH 7.4 at room temperature.

**Complex II activity.** Complex II (succinate dehydrogenase) activity was measured by complex II enzyme activity microplate assay kit (Novagen, Germany), as described by Horowitz et al. [34]. Each well in the kit has been coated with an anti-complex II monoclonal antibody which purifies the enzyme from a complex sample. After purification, the production of ubiquinol by the enzyme results in the reduction of the dye 2,6-diclorophenolindophenol with a decrease in its absorbance at 600 nm. The activity of complex II (mOD/min) is the mean of measurements obtained with enzyme minus the rate obtained without enzyme.

**Complex IV activity.** Complex IV (cytochrome-c oxidase) activity was assayed as described by Storrie and Amadden [35], using a kit purchased from Bio-vision, Milpitas, CA, USA. This method is based on monitoring the decrease in absorbance at 550 nm of ferrocytochrome-c caused by its oxidation to ferricytochrome-c where its activity was calculated in nmol cytochrome-c oxidized/min/mg protein.

## Determination of comet parameters

Briefly, fully frosted slides were pre-coated on each end with 100 μl of 0.8% agarose in phosphate-buffered saline (PBS) and left at room temperature for 20 minutes. About 10,000 cells were mixed with 70 μl of 1% low-melting-point agarose in PBS. The mixture was immediately spread onto each end of a pre-coated slide and covered with a fresh glass coverslip. After lysis, denaturation, electrophoresis, neutralization and staining, the slides were examined with a

fluorescence microscope. For each slide, 100 cells were counted at least twice. The comets were captured with an Olympus fluorescent microscope equipped with a charge-coupled device (CCD) camera, and the images were quantitatively evaluated for the percentage of DNA damage in the tail, tail length (TL) and tail moment (TM) using CASP software [36].

### Histopathological investigation

Striatum samples were kept in 10% formol saline for 24 h. Samples were washed with saline and dehydrated in serial dilutions of alcohol. Specimens were cleared in xylene and embedded in paraffin at 56˚C for 24 h. Paraffin wax tissue blocks were prepared for sectioning at a thickness of 4 μm. The sections were deparaffinized and stained by hematoxylin and eosin (H & E) (Sigma, St. Louis, MO, USA) for examination by the light microscope.

### Statistical analysis

Quantitative data were expressed as mean ± standard error of mean (SEM). One way analysis of variance (ANOVA) was used for comparing different groups. Duncan's test was applied to study statistical differences among the experimental groups. All analyses were performed using Statistical Package for the Social Sciences (SPSS) version 18 for Windows (SPSS Inc., Chicago, USA) and differences were considered statistically significant at $p < 0.05$ for all tests.

## Results

In general, 3-NP injection caused marked reduction in the total body weight, the levels of monoamines and mitochondrial complexes were accompanied with remarkable elevations in the levels of monoamine oxidases, nitric oxide, caspase-3 and lipid peroxidation in the striatum of rats. Additionally, notable instabilities in the studied behavioural parameters and histopathological alterations were observed in the 3-NP treated rats.

### Effect of 3-NP, Safr and Cands on body weight

The changes in body weight of all groups are displayed in Fig 2. As compared to the corresponding controls, the total body weight of 3-NP group was markedly ($P = 0.000$) reduced by about 20%. Though, as compared to the 3-NP group, the body weight showed remarkable elevations in rats of Safr+3-NP ($P = 0.004$) and Cands+3-NP ($P = 0.000$) groups by 11% and 13%, respectively.

### Effect of 3-NP, Safr and Cands on the passive avoidance parameters

The data of PA test parameters after 24 h and 48 h are presented (Fig 3). After one and two days, rats of 3-NP-treated group showed significant reductions ($P = 0.000$) in the latency period to step through the dark area by 87% and 91%, respectively, as compared to the CMC control group. In contrast, after 24h and 48h, rats of Safr+3-NP and Cands+3-NP groups showed significant elevations ($P = 0.000$) in the step-through latency after 24h and 48h by (6.8 & 6.3 folds) and (10.2 & 9.4 folds), respectively, as compared to the rats of 3-NP group.

### Effect of 3-NP, Safr and Cands on open field parameters

In Fig 4, Open field test parameters of all groups are reported. Rats of 3-NP group showed marked elevations in the latency period (4-folds, $P = 0.000$) and grooming frequency (2-folds, $P = 0.000$), as compared to the corresponding control group. However, the ambulation and rearing frequencies were remarkably decreased in 3-NP group by 54% ($P = 0.000$) and 80% ($P = 0.000$), respectively, in comparison to the CMC-group.

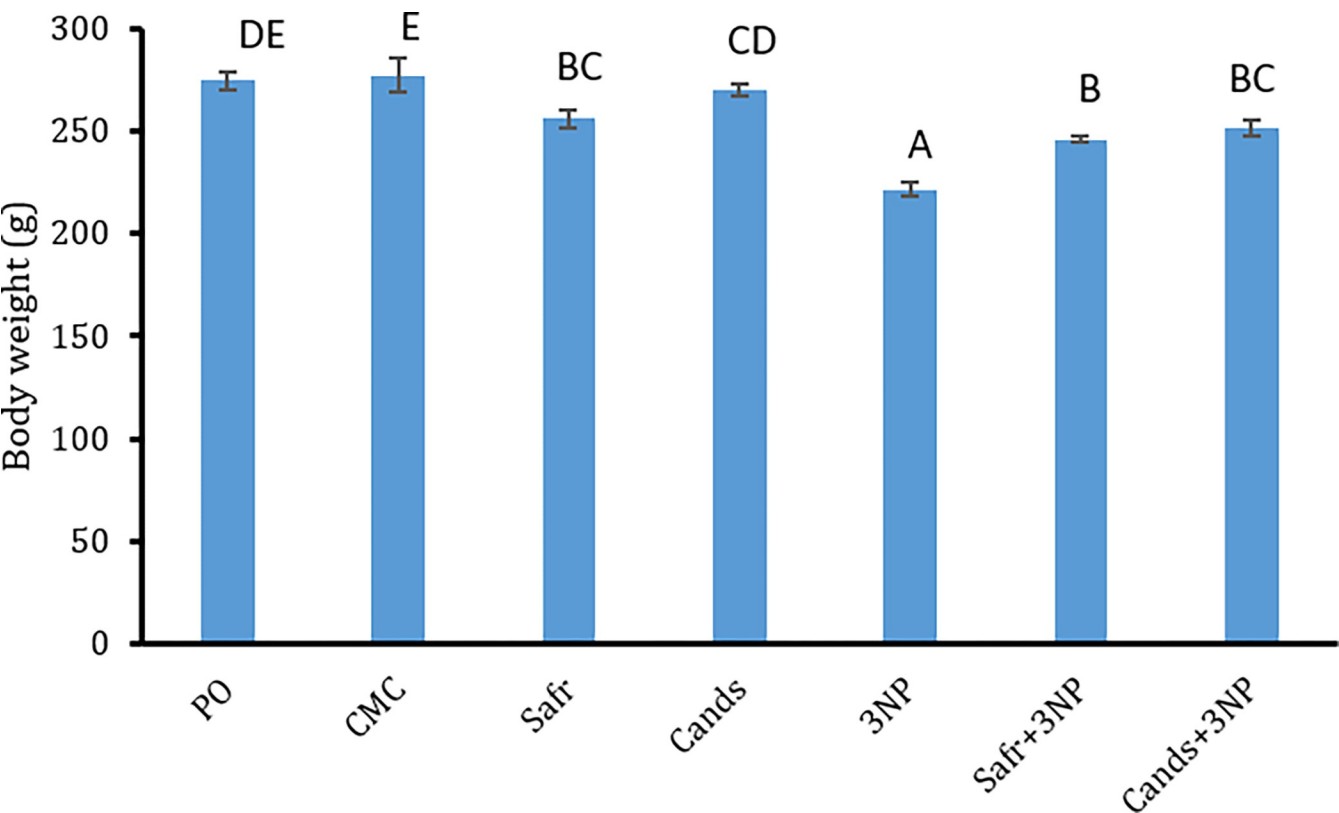

**Fig 2. Body weight changes in all the experimental groups.** Bars represent mean ± standard error. Bars marked with the same letters are insignificantly different (p>0.05), whereas those with different ones are significantly different (p<0.05).

On the other hand, Safr+3-NP- and Cands+3-NP-treated groups exhibited remarkable reductions in the latency period (53% & 44%) and grooming frequency (31%; P = 0.012 & 28%, P = 0.027). Moreover, marked elevations were observed in the ambulation frequency (1.7, P = 0.015 & 1.5 folds; P = 0.134) and rearing frequency (2.4; P = 0.001 & 2.9 folds; P = 0.000) in Safr+3-NP- and Cands+3-NP-treated groups, as compared to the 3-NP group, respectively.

### Effect of 3-NP, Safr and Cands on elevated plus maze parameters

Fig 5 demonstrates the EPM parameters in all groups. The injection of 3-NP caused a remarkable elevation in the time spent in closed arms (16%; P = 0.003) whereas marked reductions in the time spent in open arms (73%, P = 0.000), the number of entries into open (53%, P = 0.000) and closed arms (48%, P = 0.007) as well as the total number of entries (50%, P = 0.000) as compared to the CMC group.

On the contrary, rats of Safr+3-NP and Cands+3-NP groups exhibited remarkable declines in the time spent in closed arms (13 and 11%) whereas marked elevations in the time spent in open arms (4- and 3-fold), the number of entries into open (1.8- and 1.7-fold) and closed arms (1.9- and 1.8-fold) as well as the total number of entries (1.8- and 1.7-fold), respectively, as compared to 3-NP group.

### Effect of 3-NP, Safr and Cands on 5-HT, 5-HIAA, NE and DA levels in the striatum

The levels of 5-HT, 5-HIAA, NE and DA in striatum of all groups are recorded (Table 1). Rats of 3-NP group showed considerable declines in the striatal levels of 5-HT (P = 0.018), 5-HIAA

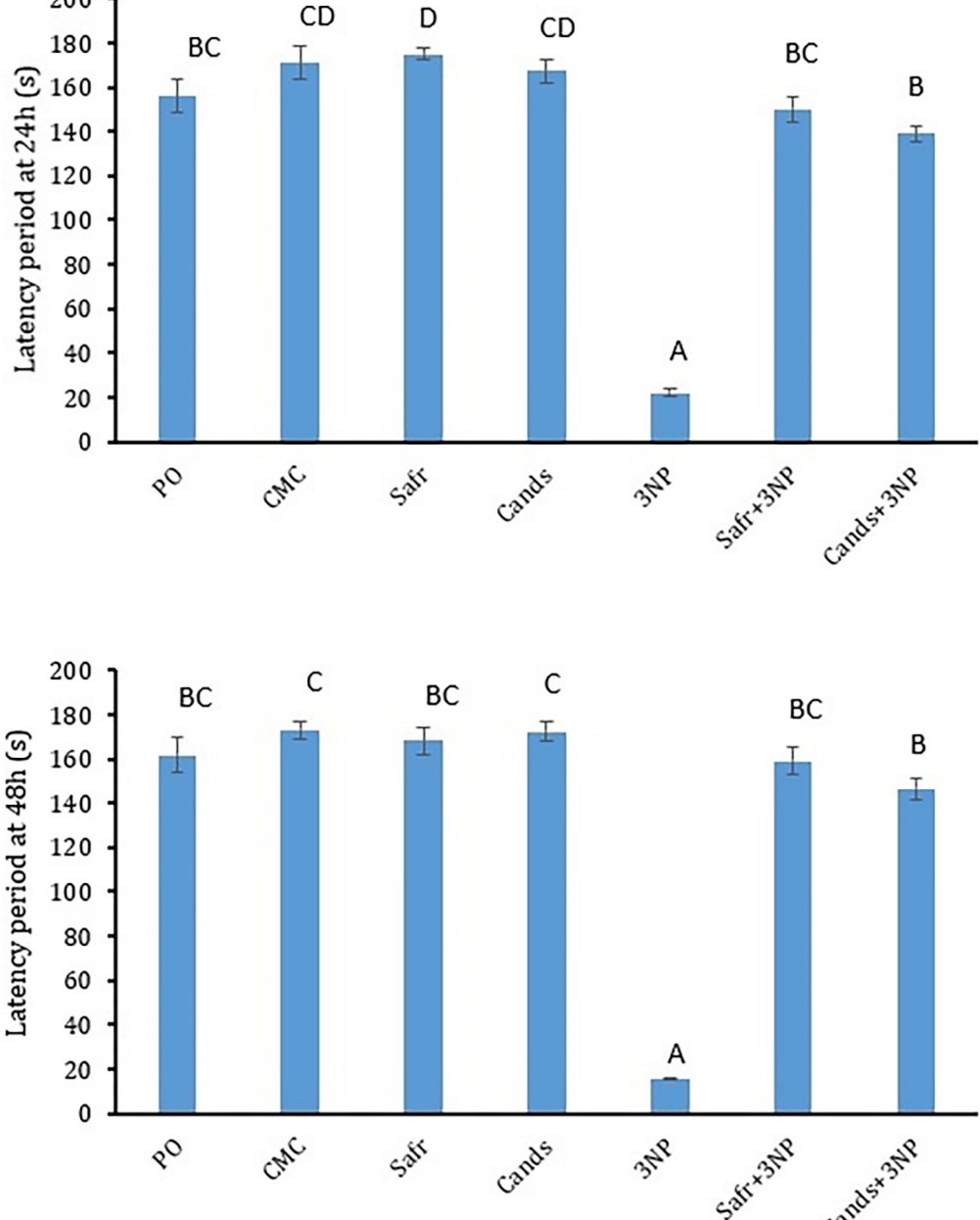

**Fig 3. The behavioral parameters in all the experimental groups, using passive avoidance test.** Bars represent mean ± standard error. Bars marked with the same letters are insignificantly different (p>0.05), whereas those with different ones are significantly different (p<0.05).

(P = 0.000), NE (P = 0.001) and DA (P = 0.000) by 37, 78, 42 and 51%, respectively, as compared to the CMC group.

On the other hand, the striatal levels of 5-HT, 5-HIAA, NE and DA in striatum of Safr +3-NP group were meaningfully higher than the 3-NP group, by 1.6 (P = 0.035), 3.8 (P = 0.000), 1.7 (P = 0.002) and 1.9-folds (P = 0.003), respectively. Similarly, Cands+3-NP group showed significant elevations in the striatal levels of 5-HT (P = 0.042), 5-HIAA

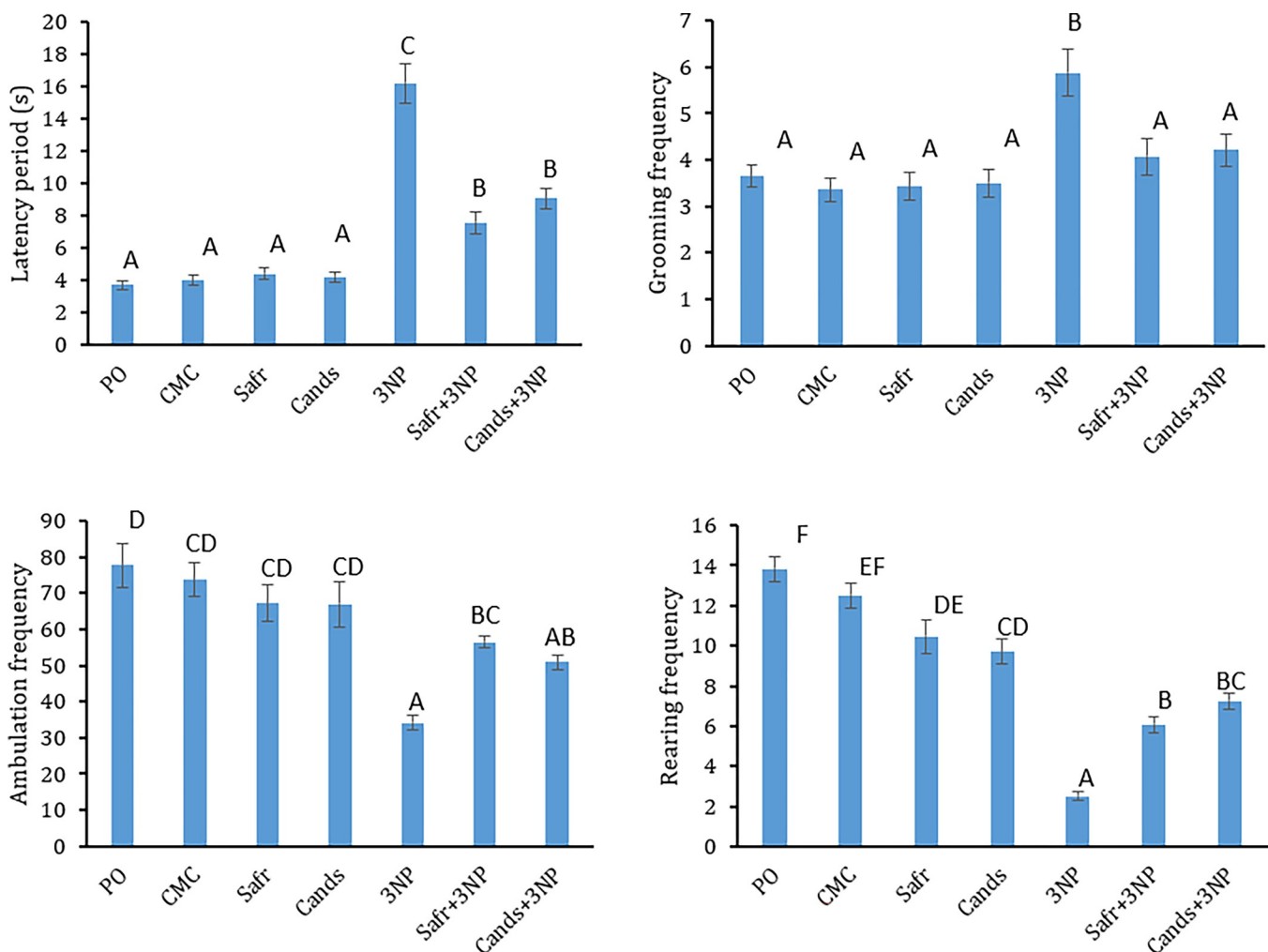

**Fig 4. The behavioral parameters in all the experimental groups, using open field test.** Bars represent mean ± standard error. Bars marked with the same letters are insignificantly different (p>0.05), whereas those with different ones are significantly different (p<0.05).

(P = 0.000), NE (P = 0.001) and DA (P = 0.007), as compared to the 3-NP group, by 1.6, 4.0, 1.7 and 1.8-folds, respectively.

## Effect of 3-NP, Safr and Cands on MAO and AChE in the striatum

The gene expression levels of MAO-A and MAO-B as well as the activities of MAO and AChE are reported (Table 2).

The striatal activity of AChE in 3-NP group showed a significant elevation by about 3-folds (P = 0.000), as compared to the CMC group. Nevertheless, rats of Safr+3-NP and Cands+3-NP groups showed marked declines in the activity of AChE by 29 (P = 0.003) and 22% (P = 0.000), respectively, as compared to the 3-NP group.

In comparison to the CMC group, the rats of 3-NP group showed a remarkable elevation in the activity of MAO (2-folds) that was associated with marked upregulations in the expression levels of MAO-A (4.5-folds) and MAO-B (7.5–folds), in the striatum. In contrast, rats of Safr +3-NP and Cands+3-NP groups showed remarkable reductions in the activity of MAO (30 and 18%) and gene expression levels of MAO-A (63 and 58%) and MAO-B (71 and 64%), respectively, as compared to the 3-NP group.

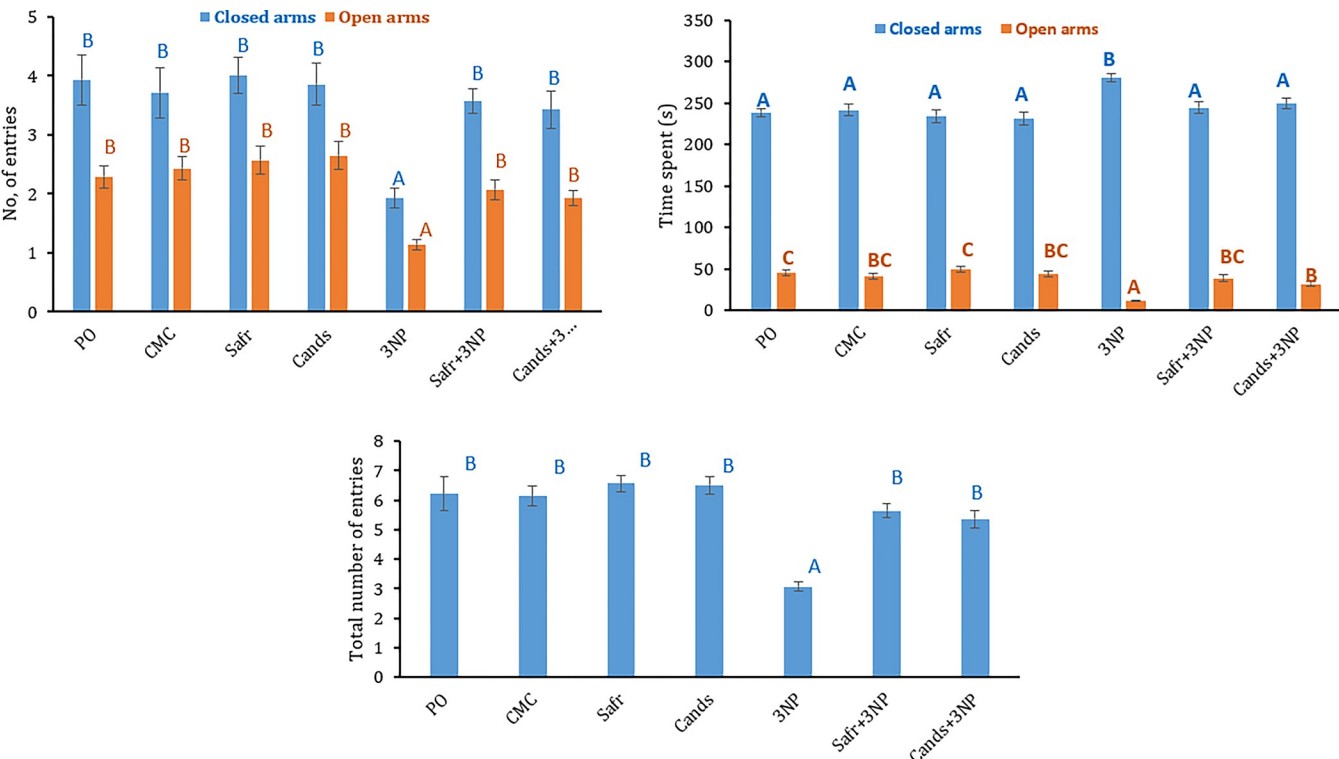

**Fig 5. The behavioral parameters in all the experimental groups, using elevated plus maze test.** Bars represent mean ± standard error. Bars marked with the same letters are insignificantly different (p>0.05), whereas those with different ones are significantly different (p<0.05).

## Effect of 3-NP, Safr and Cands on the striatum levels of NO, MDA and TAC

The striatal levels of NO, MDA and TAC in all groups are clarified (Table 3). In 3-NP group, significant elevations in the levels of NO (2.2-fold, P = 0.000) and MDA (2.9-fold, P = 0.000) were associated with a marked decline in the TAC (24%, P = 0.000), as compared to the corresponding control group.

Though, in rats of Safr+3-NP and Cands+3-NP groups significant declines in the levels of NO (53 and 51%) and MDA (59 and 54%) were accompanied with a remarkable elevation in the TAC (28, P = 0.000 and 21%, P = 0.014), respectively, as compared to 3-NP-induced rats.

**Table 1. Effect of Safr and Cands on the levels of 5-hydroxytryptamine (5-HT, µg/g tissue), 5-hydroxyindoleacetic acid (5-HIAA, µg/g tissue), norepinephrine (NE, µg/g tissue) and dopamine (DA, µg/g tissue) in striatum of 3-NP-treated rats.**

| Parameter | Control groups | | Experimental groups | | | | | Effect of treatment |
|---|---|---|---|---|---|---|---|---|
| | PO | CMC | Safr | Cands | 3-NP | Safr + 3-NP | Cands + 3-NP | |
| 5-HT | 0.34±0.03$^B$ | 0.35±0.03$^B$ | 0.35±0.03$^B$ | 0.36 ±0.04$^B$ | 0.21 ±0.02$^A$ | 0.34±0.03$^B$ | 0.34±0 .02$^B$ | $F_{6,63}$ = 3.29, P<0.007 |
| 5-HIAA | 0.21±0.01$^B$ | 0.24±0.01$^B$ | 0.25±0.03$^B$ | 0.20±0.02$^B$ | 0.051±0.01$^A$ | 0.20±0.02$^B$ | 0.21±0.02$^B$ | $F_{6,63}$ = 15.59, P<0.000 |
| NE | 0.86±0.04$^B$ | 0.87±0.06$^B$ | 0.89±0.08$^B$ | 0.89±0.07$^B$ | 0.50±0.04$^A$ | 0.85±0.05$^B$ | 0.85± 0.05$^B$ | $F_{6,63}$ = 5.87, P<0.000 |
| DA | 0.87±0.09$^B$ | 0.90±0.05$^B$ | 0.86±0.04$^B$ | 0.87±0.08$^B$ | 0.44±0.04$^A$ | 0.82±0.08$^B$ | 0.79± 0.07$^B$ | $F_{6,63}$ = 6.04, P<0.000 |

In the same row, values marked with the same superscript CAPITAL letters are insignificantly different (P>0.05), whereas those with different ones are significantly different (P<0.05). P<0.000: Represents significant effect of treatment.

**Cands**: Candesartan, **CMC**: Carboxymethyl cellulose, **PO**: Paraffin oil. **Safr**: Safranal, **and 3-NP**, 3-nitropropionic acid.

Data is displayed as mean ± standard error (SE).

**Table 2. Effect of Safr and Cands on the gene expression level of monoamine oxidase (MAO-A, fold change) and (MAO-B, fold change) as well as the activities of monoamine oxidase (MAO, µM 4-HQ /g tissue) and acetylcholinesterase (AChE, U/mg protein) in striatum of 3-NP-treated rats.**

| Parameter | Control groups | | Treated groups | | | | | Effect of treatment |
|---|---|---|---|---|---|---|---|---|
| | PO | CMC | Safr | Cands | 3-NP | Safr + 3-NP | Cands + 3-NP | |
| MAO-A | 1.02 ±0.01$^A$ | 1.02±0.01$^A$ | 1.02±0.01$^A$ | 1.01±0.02$^A$ | 4.61±0.41$^C$ | 1.69±0.13$^B$ | 1.92±0 .03$^B$ | $F_{6,63}$ = 63.66, P<0.000 |
| MAO-B | 1.01 ±.00$^A$ | 1.00±.00$^A$ | 1.01±0.00$^A$ | 1.01±0.01$^A$ | 7.54±0.02$^D$ | 2.18±0.13$^B$ | 2.75±0 .09$^C$ | $F_{6,63}$ = 566.96, P<0.000 |
| MAO activity | 12.85 ±1.14$^A$ | 12.62±0.86$^A$ | 13.53 ±0.63$^A$ | 16.57±0.21$^B$ | 25.20±.14.00$^D$ | 17.55±0.49$^B$ | 20.79±0.39$^C$ | $F_{6, 63}$ = 41.99, P<0.000 |
| ACHE activity | 61.00 ±5.10$^A$ | 61.40±4.90$^A$ | 78.20 ±6.40$^B$ | 96.80 ±6.00$^B$ | 185.00±8.60$^D$ | 130.90±7.70$^C$ | 143.60± 9.30$^C$ | $F_{6, 63}$ = 44.15, P<0.000 |

In the same row, values marked with the same superscript CAPITAL letters are insignificantly different (P>0.05), whereas those with different ones are significantly different (P<0.05). P<0.000: Represents significant effect of treatment.

**Cands**: Candesartan, **CMC**: Carboxymethyl cellulose, **HQ**:4-hydroxyquinoline, **PO**: Paraffin oil. **Safr**: Safranal, **and 3-NP**, 3-nitropropionic acid.

Data is displayed as mean ± standard error (SE).

## Effect of 3-NP, Safr and Cands on the activities of complexes I, II and IV in the striatum

The activities of the mitochondrial complexes I, II and IV were displayed (Table 4). In comparison to the corresponding control group, the activities of mitochondrial complexes I, II and IV in stratum of 3-NP group were significantly (P = 0.000) declined by 70, 65 and 56%, respectively.

On the other hand, the levels of complexes I, II and IV in the striatum of Safr+3-NP and Cands+3-NP groups were remarkably (P = 0.000) greater than in 3-NP group (2.8 and 2.6-fold), (2.3 and 1.9-fold) along with (1.7 and 1.5-fold), respectively.

## Effect of 3-NP, Safr and Cands on Casp-3, Fas-L and iNOS protein levels in the striatum

In Table 4, the protein levels of Casp-3, Fas-L and iNOS were presented. The rats of 3-NP group showed a significant (P = 0.000) elevation in the protein levels of Casp-3, Fas-L and iNOS by (5.8, 5.1 and 6.3-fold), in relation to the corresponding control group. On the other hand, in striatum of Safr+3-NP and Cands+3-NP groups, marked (P = 0.000) reductions in the protein levels of Casp-3 (65 and 66%), (61 and 58%) and iNOS (61 and 58%) were reported as compared to the 3-NP group, respectively.

**Table 3. Effect of Safr and Cands on the levels of malondialdehyde (MDA, nmol/g tissue), nitric oxide (NO, nmol/g tissue), and total antioxidant capacity (TAC, mmol/mg protein) in striatum of 3-NP-treated rats.**

| Parameter | Control groups | | Experimental groups | | | | | Effect of treatment |
|---|---|---|---|---|---|---|---|---|
| | PO | CMC | Safr | Cands | 3-NP | Safr + 3-NP | Cands + 3-NP | |
| MDA | 94.45±9.23$^A$ | 90.60±8.73$^A$ | 88.85±2.41$^A$ | 87.76±4.03$^A$ | 263.09±20.54$^B$ | 107.54±3.99$^A$ | 120.16±11.79$^A$ | $F_{6,63}$ = 37.02, P<0.000 |
| NO | 650.50±19.10$^A$ | 642.70±31.90$^A$ | 650.40±55.70$^A$ | 652.30±50.30$^A$ | 1404.10±37.30$^B$ | 666.00±65.00$^A$ | 685.50±60.80$^A$ | $F_{6,63}$ = 34.17, P<0.000 |
| TAC | 80.90±1.50$^B$ | 81.30±4.20$^B$ | 80.20±0.90$^B$ | 80.50±5.80$^B$ | 61.60 ±0.80$^A$ | 78.90±1.00$^B$ | 74.70±1.30$^B$ | $F_{6,63}$ = 7.31, P<0.000 |

In the same row, values marked with the same superscript CAPITAL letters are insignificantly different (P>0.05), whereas those with different ones are significantly different (P<0.05). P<0.000: Represents significant effect of treatment.

**Cands**: Candesartan, **CMC**: Carboxymethyl cellulose, **PO**: Paraffin oil. **Safr**: Safranal, **and 3-NP**, 3-nitropropionic acid.

Data is displayed as mean ± standard error (SE).

**Table 4. Effect of Safr and Cands on the levels of complex I (U/mg protein), complex II (U/mg protein), complex IV (U/mg protein), caspase-3 (Casp-3, µg/mg protein), Fas ligand (FasL, µg/mg protein) and inducible nitric oxide synthase (iNOS, µg/mg protein) in striatum of 3-NP-treated rats.**

| Parameter | Control groups | | Experimental groups | | | | | Effect of treatment |
|---|---|---|---|---|---|---|---|---|
| | PO | CMC | Safr | Cands | 3-NP | Safr + 3-NP | Cands + 3-NP | |
| Complex I | 80.82 ±2.91$^C$ | 80.82±2.91$^C$ | 80.78±2.58$^C$ | 81.47±2.27$^C$ | 24.14±1.15$^A$ | 66.09±1.09$^B$ | 62.58± 1.00$^B$ | $F_{6,63}$ = 95.03, P<0.000 |
| Complex II | 49 ±.1.63$^C$ | 49±1.63$^C$ | 49.31±0.78$^C$ | 49.15±2.04$^C$ | 16.92±0.85$^A$ | 38.06±0.76$^B$ | 32.78± 1.06$^B$ | $F_{6,63}$ = 85.52, P<0.000 |
| Complex IV | 26.88 ±.95$^C$ | 26.88±0.95$^C$ | 27.06±1.15$^C$ | 26.93±2.27$^C$ | 11.78±0.54$^A$ | 19.99±0.39$^B$ | 18.03± 1.89$^B$ | $F_{6,63}$ = 23.85, P<0.000 |
| Casp-3 | 1.00 ±0.00$^A$ | 1.01±0.01$^A$ | 1.00±0.00$^A$ | 1.01±0.01$^A$ | 5.80±0.48$^C$ | 2.06±0.56$^B$ | 1.97±0.44$^B$ | $F_{6,63}$ = 174.59, P<0.000 |
| FasL | 1.01 ±0.00$^A$ | 1.01±0.00$^A$ | 1.00±0.01$^A$ | 1.01±0.02$^A$ | 5.17±0.18$^C$ | 2.03±0.04$^B$ | 2.18±0.05$^B$ | $F_{6,63}$ = 455.51, P<0.000 |
| iNOS | 1.01±0.00$^A$ | 1.01±0.00$^A$ | 1.04±0.01$^A$ | 1.00±0.00$^A$ | 6.38±0.11$^C$ | 2.48±0.09$^B$ | 2.70±0.04$^B$ | $F_{6,63}$ = 1230.8, P<0.000 |

In the same row, values marked with the same superscript CAPITAL letters are insignificantly different (P>0.05), whereas those with different ones are significantly different (P<0.05).

P<0.000: Represents significant effect of treatment.

**Cands**: Candesartan, **CMC**: Carboxymethyl cellulose, **PO**: Paraffin oil. **Safr**: Safranal, and **3-NP**, 3-nitropropionic acid.

Data is displayed as mean ± standard error (SE).

## Effect of 3-NP, Safr and Cands on the percent of DNA damage in the striatum

Comet assay parameters in striatal cells of all groups were analyzed (Table 5). In 3-NP group, the % DNA damage in tail (2-fold), TL (2-fold) and TM (4-fold) were significantly (P = 0.000) elevated, as compared to the corresponding control group. However, both Safr+3-NP and Cands+3-NP groups exhibited remarkable (P = 0.000) declines in the % DNA damage (19 and 14%), TL (24 and 20%) and TM (39 and 27%), in comparison to the 3-NP group.

## Effect of 3-NP, Safr and Cands on histopathological alterations in the striatum

The striatum area in the brain tissue of all the experimental groups were microscopically investigated (Figs 6–9). In the control groups, photomicrograph of striatum showed healthy architecture consisting of uniform, well-organized striatal cells with basophilic cytosol and vesicular nuclei. In addition, perineuronal spaces were observed around neurons (Fig 6). The striatum tissue of mice treated with safranal alone or candesartan alone showed similar appearance to that of the controls without any histopathological alterations (Fig 7). On the other hand, the 3-NP group showed remarkable alterations in the histological structure of striatum such as

**Table 5. Effect of Safr and Cands on the levels of % DNA damage in tail (arbitrary unit), tail length (TM, Mm) and tail moment (TM, arbitrary unit) in striatum of 3-NP-treated rats.**

| Parameter | Control groups | | Experimental groups | | | | | Effect of treatment |
|---|---|---|---|---|---|---|---|---|
| | PO | CMC | Safr | Cands | 3-NP | Safr + 3-NP | Cands + 3-NP | |
| %DNA | 2.06±0.11$^A$ | 2.09±0.09$^A$ | 2.17±0.09$^A$ | 2.27±0.06$^A$ | 4.11±0.09$^C$ | 3.30±0.03$^B$ | 3.55± 0.03$^B$ | $F_{6,63}$ = 116.6, P<0.000 |
| TL | 2.07±0.06$^A$ | 1.99±0.03$^A$ | 2.21±0.04$^A$ | 2.19±0.17$^A$ | 3.98±0.06$^C$ | 3.03±0.09$^B$ | 3.19± 0.18$^B$ | $F_{6,63}$ = 49.43, P<0.000 |
| TM | 4.15±0.14$^A$ | 4.22±0.22$^A$ | 4.78±0.13$^A$ | 4.54±0.06$^A$ | 16.24±0.37$^D$ | 9.86±0.18$^B$ | 11.89± 0.52$^C$ | $F_{6,63}$ = 301.9, P<0.000 |

In the same row, values marked with the same superscript CAPITAL letters are insignificantly different (P>0.05), whereas those with different ones are significantly different (P<0.05).

P<0.000: Represents significant effect of treatment.

**Cands**: Candesartan, **CMC**: Carboxymethyl cellulose, **PO**: Paraffin oil. **Safr**: Safranal, **and 3-NP**, 3-nitropropionic acid.

Data is displayed as mean ± standard error (SE).

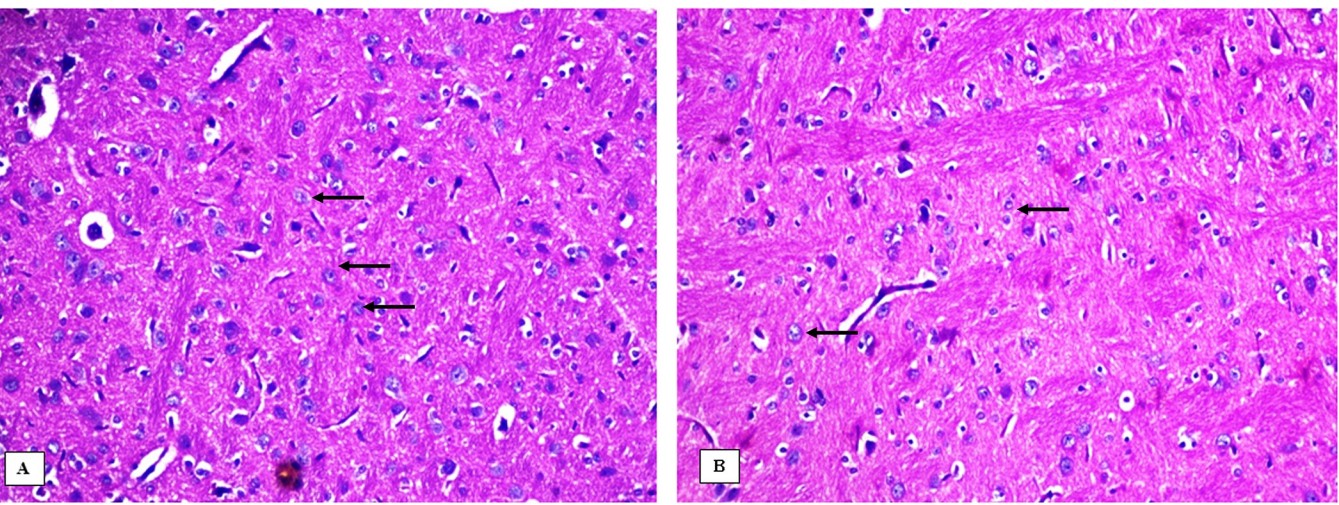

**Fig 6.** Photomicrograph of the striatum of the negative control groups A) PO-treated and B) CMC-treated, showing the normal histological structure of the neurons in striatum (H&E 400x). Black arrows represent normal cells.

nuclear pyknosis of the degenerated neurons with gliosis satellatosis and neuronophagia (Fig 8). In Safr+3-NP and Cands+3-NP groups, striatal tissue restored the normal histological structure (Fig 9).

## Discussion

Our findings indicated that Safr (50mg/kg) and Cands (1mg/kg) may prevent or alleviate the progression HD and its associated impairments induced by the daily intraperitoneal injection of 3-NP (20 mg/kg), throughout the experimental duration (9 days). Our results are in accordance with the recent study by Fotoohi et al [21], regarding the beneficial effects of Safr (0.75, 1.5 and 3.0 mg/kg) on the movement defects induced by 3-NP (10 mg/kg), during the two weeks of experiment. However, our study was designed to investigate the possible mechanisms

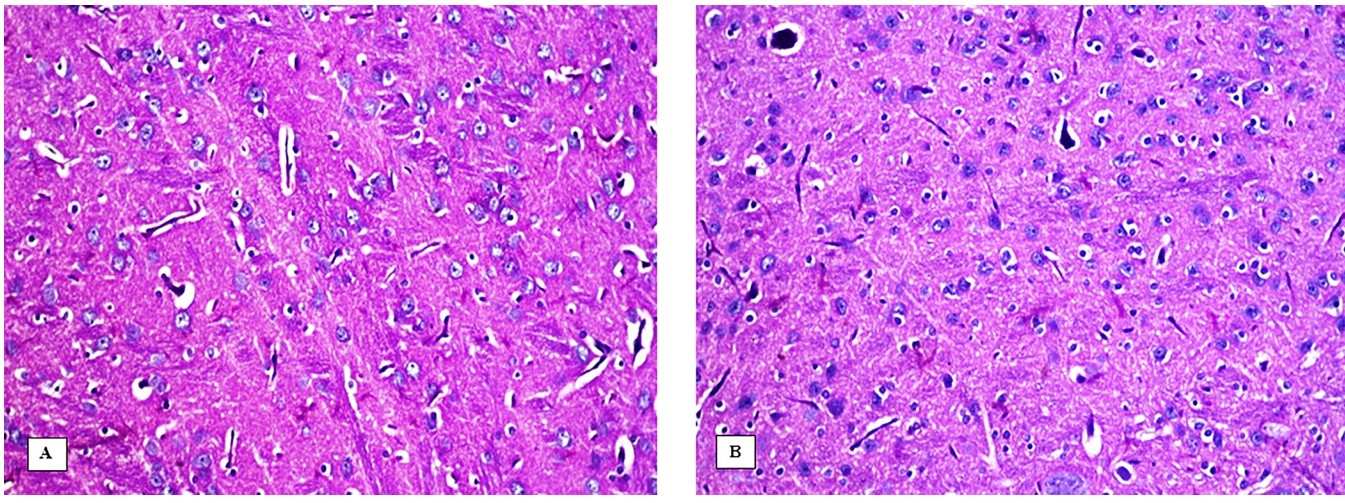

**Fig 7.** Photomicrograph of the striatum of A) Safr-treated and B) Cands-treated groups showing the no histopathological changes in the structure of the neurons in striatum (H&E 400x).

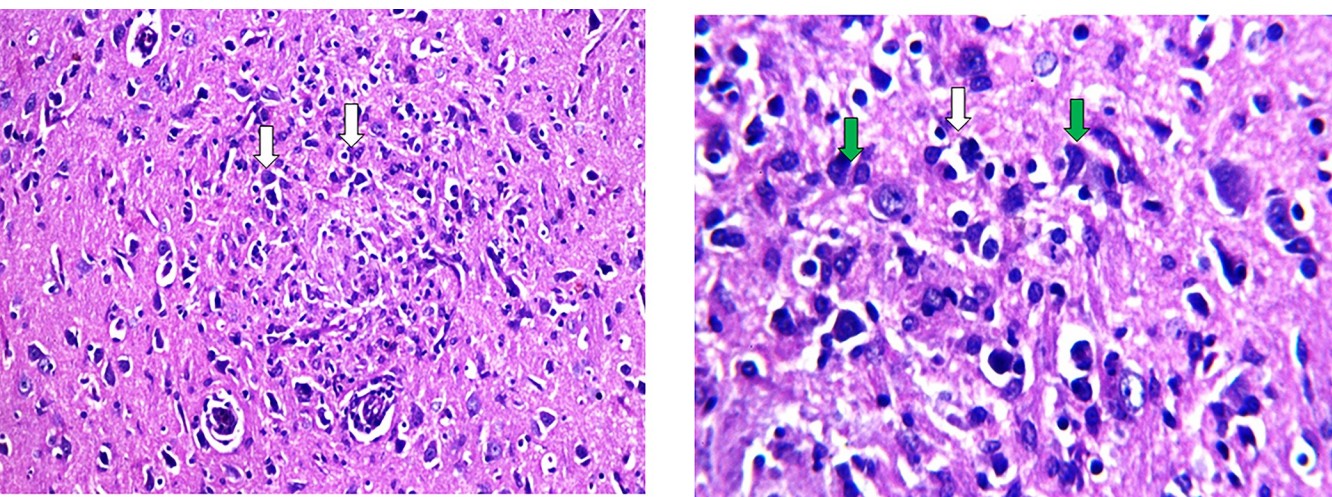

**Fig 8. Photomicrograph of the striatum of 3-NP-ijnected group showing nuclear pyknosis of the degenerated neurons (white arrow) with gliosis satellatosis (green arrow) and neuronophagia (H&E 400x & 1000x).**

that might explain the potential protective roles of Safr and Cands against 3-NP-induced alterations. Accordingly, in the current study, the changes in behavioral, biochemical (oxidative stress biomarkers, inflammatory mediators, apoptotic proteins, and mitochondrial complexes) as well as histopathological investigations were monitored.

The reported weight reduction after 3-NP injection, in the present work, is in accordance with a previous study by Gao *et al.* [37]. 3-NP injection can hinder the action of succinate dehydrogenase (SDH) in tricarboxylic acid cycle, leading to ATP depletion [38], leading to excessive metabolism of the stored glycogen and fats and subsequent weight loss [39]. Moreover, 3-NP may reduce appetite through inducing striatal injuries [40].

According to **OFT**, 3-NP altered ambulation, grooming and rearing frequencies, indicating impaired locomotor activity. Results of **EPM** and **PA** tests revealed that 3-NP also caused anxiety, altered motor activity as well as impaired memory. The decreased time spent in open

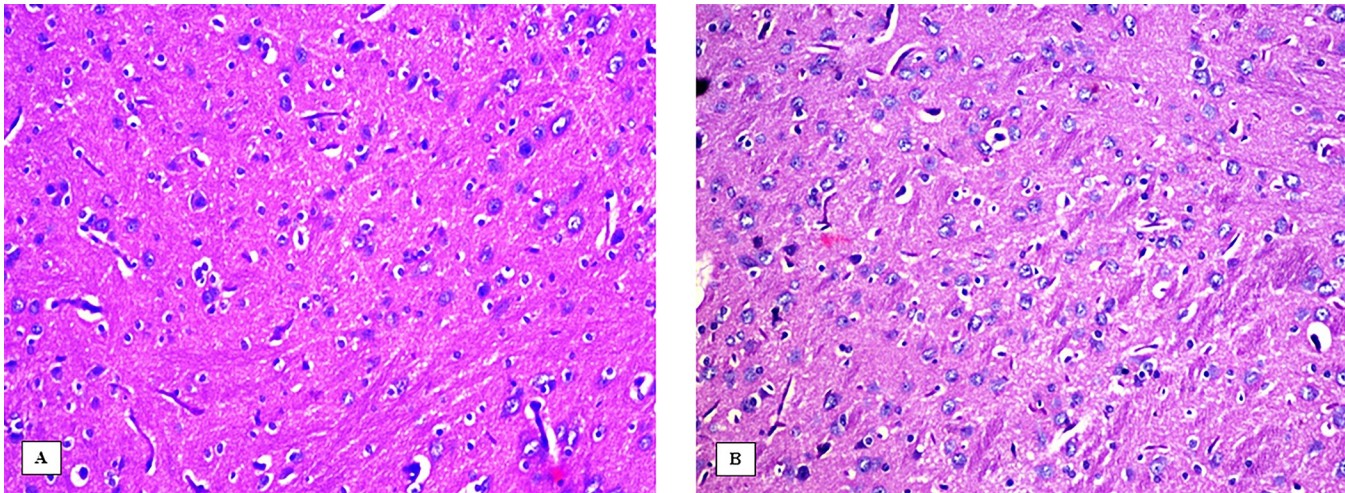

**Fig 9.** Photomicrograph of the striatum of A) Safr+3-NP and B) Cands+3-NP groups showing no histopathological changes in the structure of the neurons in striatum (H&E 400x).

arms, number of total and close arm entries by 3-NP rats may reflect the increased anxiety while reduced spontaneous motor activity [41]. Several recent studies reported similar findings in 3-NP treated rats [42–44]. In the brain, stress can boost angiotensin II (Ang II) levels [45], causing stimulation of Ang II receptor that increases Adrenocorticotropic Hormone (ACTH) secretion from the pituitary gland and subsequent release of corticosterone [46]. In rodents, corticosterone can adversely affect cognitive memory [47]. The observed behavioral disturbances were associated with remarkable declines in the striatal neurotransmitter levels in 3-NP treated rats. Mahdi *et al* [48] also reported that 3-NP altered the levels of DA, 5-HT and NE in rats. These findings can be explained by the reported increase in the striatal MAO activity, in the 3-NP group, that is responsible for the breakdown of monoamines in the brain [49]. MAO has two isoenzymes MAO-A (in catecholaminergic neurons) and MAO-B (in glial cells) [50]. MAO catalyzes oxidative deamination of catecholamines (such as DA and NE) and 5-HT [51]. The current results showed that 3-NP injection significantly upregulated the mRNA levels of MAO-A and -B.

AChE plays a key role in cholinergic transmission. Thus, the increased activity of AChE can reduce acetylcholine (ACh) level, leading to altered learning and memory activities [52], as indicated in the preset study. Moslemi *et al.* [53] reported similar findings in striatum, cortex and hippocampus areas of male and female rats.

Oxidative stress may play a major role in the 3-NP induced neurotoxicity, as manifested by recorded elevation in MDA level whereas a marked decline in TAC level, in striatum. MDA is a lipid peroxidation biomarker that is implicated in the cellular dysfunction in the brain under stress and neurodegenerative diseases [54]. Mahdy *et al.* [55] also detected that 3-NP can induce oxidative damage in the striatum of rats, through enhancing ROS generation.

The current data showed significant suppression in the mitochondrial complexes I, II and IV, indicating 3-NP-induced mitochondrial dysfunction. 3-NP can irreversibly bind to complex II, leading to interruption of the respiratory chain [25], causing subsequent ROS excessive production. Moreover, 3-NP may inhibit the action of complexes I and IV by via S-nitrosation of their subunits [56].

3-NP injection caused a marked increase in NO level, indicating 3-NP induced neuroinflammation. This can be linked to the recorded increase in iNOS activity. NO can impair the spatial memory via promoting neuronal cell death [57], as ensured by the results of elevated plus maze. Renin–angiotensin system (RAS) is involved in the onset and progression of inflammation as well as oxidative damage in central nervous system (CNS), through boosting the Ang II level [58,59]. Noteworthy that the activation of RAS can upregulate inflammatory mediators that contribute to neurodegenerative disorders [60]. In the same way, Ibrahim and Rasheed [43] recorded that 3-NP injection markedly upregulated iNOS and NO levels in the striatum of rats.

3-NP upregulated Casp-3 activity in striatum indicating 3-NP induced apoptosis. This could be linked to the increased level of FasL. FasL activates caspase cascade via binding to a tumor necrosis factor receptor (Fas). Casp-3 is a member of the caspase family that regulates the apoptotic process [61]. These findings were manifested by the reported DNA fragmentation. Similarly, Abdelfattah *et al.* [2] reported that 3-NP caused marked elevations in Casp-3 activity and Bcl-2-associated X protein (Bax) level, in the striatum. In addition, histopathological investigation showed nuclear pyknosis of neurons, indicating irreversible signs of apoptosis. Similar histopathological alterations were reported by Tasset *et al* [62]. They attributed that to the 3-NP induced excessive generation of ROS.

Safr is a monoterpene aldehyde that represents the major volatile component of safron. Safr gives saffron its characteristic aroma [63]. The present data showed that Safr prevented the 3-NP-induced body weight loss. Similar results were obtained by Shahat *et al.* [22], in

hyperthyroidism rat model after safr administration. They suggested that the sulfhydryl group in safr may interrupt thyroid hormone biosynthesis through decreasing iodide uptake, hydrogen peroxide level, and thyroid peroxidase activity. Noteworthy that the thyroid hormones are responsible for accelerating basal metabolic rate and protein wasting of the body.

Safr also prevented the 3-NP-induced reduction in the striatal levels of neurotransmitter. Safr has immense antioxidative properties through scavenging free radicals [64]. This was ensured by the reported decline in the MDA level associated with a remarkable elevation in TAC levels. The free radical scavenging activity of safr can hinder the ROS-induced impairments in the neurotransmission [65]. Moreover, safr can inhibit the reuptake of dopamine, serotonin and norepinephrine via direct binding to their receptors [66]. In addition, safr caused significant reduction in the MAO activity, preventing the degradation of monoamines. The induced decline in MAO activity can be linked to the free radical scavenging action of safr. Noteworthy that MAO is hydrogen peroxide ($H_2O_2$)-dependent enzyme [22]. Accordingly, the reported improvement in learning and memory activities, after safr treatment can be linked to the enhanced dopamine and serotonin levels as well as the improved blood flow to the brain [67]. In the present study, Safr administration remarkably reduced anxiety as confirmed by the results of EPM test. This may be linked to the reported Safr-induced elevation in the striatal levels of monoamines including serotonin, that is responsible for the mood improvement [68]. In the same line, **Pontifex et al [69]** reported that saffron extract had antidepressant and anti-anxiety properties. They attributed that to the improvement in anxiety-related behaviour were linked with the ability of saffron to affect the brain through inducing gut microbial shifts via gut–brain axis interactions. In addition, the spatial memory may be enhanced through Safr-enhanced AChE activity [70], as reported in the present study. In addition, the results of PA test revealed that spatial memory was improved by administration of Safr. This can be inked to the ability of Safr to reduce the apoptotic enzyme Casp-3 activity and MDA level, leading to reduced neuronal loss in the striatum. In the current study, Safr significantly improved locomotor activity as revealed by the results of OFT. These findings may be attributed to the ability of Safr to increase the striatal levels of norepinephrine and dopamine which are highly involved in motor control [71].

As no histopathological damage was observed in the striatum of Safr-treated rats. Similarly, several previous studies illustrated the ameliorative role of Safr against oxidative damage-induced destructions in the hippocampus [72], as well as ischemia-reperfusion-enhanced damages in the cerebral cortex [73].

In addition, Safr markedly reduced the Cas-3 and FasL levels, indicating its anti-apoptotic properties, in the current data. This can be linked to the antioxidative properties of Safr. In addition, Safr can inhibit the release of cytochrome c, leading to inactivation of Cas-3 [22].

The iNOS level significantly declined in the Safr-treated rats before 3-NP. This may explain the reduced levels of NO, reflecting its anti-inflammatory action. In the same manner, Bukhari et al. [74] reported that Safr treatment caused remarkable decline in iNOS and peroxynitrite levels, besides restoration of mitochondrial function and DNA integrity. Koul and Abraham [75] reported that Safr protected the bone marrow of mice against the DNA damaging effect of gamma radiation. They attributed that to its antigenotoxic properties. Safr may protect DNA from harmful damage through external binding or intercalative styles [76].

The current results showed that Cands hindered 3-NP-induced body weight loss. Chen et al. [77] also reported that an oral dose of 5 mg/kg/day prevented body weight loss in rats under chronic stress. Furthermore, Cands treatment improved motor activity and exploratory behavior as indicated by the results of OFT. This can be also linked to the reported elevation in NE and DA levels in striatum after treatment with Cands. In addition, Cands enhanced spontaneous motor activity and reduced 3-NP-fear drive, as indicated by EPM results, indicating

anti-anxiety action of Cands. This can be due to the ability of increase the striatal serotonin levels, in the present data. PA results revealed that Cands improved the spatial memory. Braszko et al. [78] also recorded that Cands prevented the stress-induced memory impairment in rats. In the same manner, Thakur et al. [60] recorded that administration of Cands at doses of 3 and 5 mg/kg remarkably improved the observed changes induced by haloperidol in rearing and grooming behavior as well as retention time in the open arms of EPM. The improved performance of rats in the behavioral tests after treatment with Cands may be linked to enhanced cholinergic system as confirmed by the reported reduction in AChE activity and subsequent increase in acetylcholine. Previous studies showed that Ang II receptor blockers can improve cognitive performance via preventing Ang II inhibitory action on the release of acetylcholine from entorhinal cortex [79].

One of the proposed mechanisms of action of Cands can be through preventing oxidative damage. Cands administration remarkably reduced the 3-NP induced elevation in the MAO activity. This could be linked to the reported Cands-induced downregulation of MAO. As MAO is responsible for $H_2O_2$ generation [80]. Thus, Cands can reduce $H_2O_2$ generation leading to decreasing the 3-NP-induced oxidative stress. This may explain reported reduction in MDA levels whereas increased TAC levels in striatum, after treatment with Cands. Similar findings were reported by Tota *et al* [81], in the brain of mice after an intraperitoneal injection with 0.1 mg/kg in mice. They linked that to the ability of Cands to scavenge free radicals. Cands has the ability to inhibit the ROS production by increasing the glutathione (GSH) content and normalizing the redox status [82]. Beţiu *et al* [83] stated that Cands at doses of 0.1–0.3mg/kg for 7 days remarkably alleviated mitochondrial electron transport system dysfunction. This was ensured by the improved Complexes I, II and IV, after treatment with Cands, in the present study. Moreover, nicotinamide adenine dinucleotide phosphate (NADPH) oxidase induces oxidative damage through enhancing superoxide anion generation [84], after being activated by binding of Ang II to AT1 receptors [85]. Thus, Cands can also indirectly hinder 3-NP-induced oxidative damage via blocking Ang II receptors [78].

Cands treatment also prevented the 3-NP-induced elevations in striatal level of NO, reflecting its anti-inflammatory action. This finding can be ascribed to the reported reduction in iNOS activity after treatment with cands. Likewise, Benicky *et al*. [86] described that an oral dose of 1 mg/kg of candesartan markedly reduced LPS-induced inflammation in the brain. Thakur et al. [60] detected that Cands has anti-inflammatory actions against haloperidol-induced inflammation in rats. They linked that to its ability to downregulate the expression of proinflammatory cytokines such as IL-1β and pro-inflammatory mediators as tumor necrosis factor-alfa (TNF-α).

Cands also caused marked decline in the Cas-3 activity and FasL level, indicating anti-apoptotic properties. Thakur *et al*. [60] reported that cands as other ARB have antioxidative and anti-apoptotic effects. ARBs such as Cands have been reported to show their neuroprotective activity through regulating key proteins such as neurotrophic factors and dopamine transporter [87]. These proteins are highly involved in the progressive degeneration of dopaminergic system as reported in many neurodegenerative diseases [88]. Accordingly, cands can penetrate blood brain barrier (BBB) and modulate 3-NP-induced HD symptoms via inhibiting the stimulating action of Ang II on dopamine receptors, preventing neuronal cell death [89].

Collectively, the current study demonstrated that the 3-NP-induced changes in the levels of neurotransmitters, oxidative stress indices and inflammatory mediators led to a decrease in the cognitive and locomotor activities. Interestingly, co-administration of Safr or Cands along with 3-NP was found to minimize or even prevent the biochemical, behavioral and histological disturbances in the experimented rats. Safr was mostly more effective than Cands regarding behavioral tests, MAO, AChE, TAC and DNA damage. Consequently, this research suggests

that Safr and Cands exert neuroprotective effects in a rat model of 3-NP-induced HD and hence might have potential as drug candidates for the treatment of HD.

## Author Contributions

**Conceptualization:** Nagwa Ibrahim Shehata, Sherine Maher Rizk.

**Data curation:** Dina Mohamed Abd EL-Salam.

**Methodology:** Nagwa Ibrahim Shehata, Roqaya Mahmoud Hussein.

**Supervision:** Sherine Maher Rizk.

**Writing – original draft:** Sherine Maher Rizk.

**Writing – review & editing:** Dina Mohamed Abd EL-Salam, Sherine Maher Rizk.

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
