## [Decision Letter · Decision Letter 0]

20 Apr 2023

PONE-D-23-06048Effect of safranal or candesartan on 3-nitropropionicacid-induced biochemical, behavioral and histological alterations in a rat model of Huntington’s diseasePLOS ONE

Dear Dr. Risk,

Thank you for submitting your manuscript to PLOS ONE. After careful consideration, we feel that it has merit but does not fully meet PLOS ONE’s publication criteria as it currently stands. Therefore, we invite you to submit a revised version of the manuscript that addresses the points raised during the review process.

We look forward to receiving your revised manuscript.

Kind regards,

Yasmina Abd‐Elhakim

Academic Editor

PLOS ONE

Journal Requirements:

Reviewers' comments:

Reviewer's Responses to Questions

**Comments to the Author**

1. Is the manuscript technically sound, and do the data support the conclusions?

Reviewer #1: Yes

Reviewer #2: Partly

2. Has the statistical analysis been performed appropriately and rigorously? 

Reviewer #1: Yes

Reviewer #2: Yes

3. Have the authors made all data underlying the findings in their manuscript fully available?

Reviewer #1: Yes

Reviewer #2: Yes

4. Is the manuscript presented in an intelligible fashion and written in standard English?

Reviewer #1: Yes

Reviewer #2: Yes

5. Review Comments to the Author

**Reviewer #1: **The manuscript “Effect of safranal or candesartan on 3-nitropropionicacid-induced biochemical, behavioral and histological alterations in a rat model of Huntington's disease” clarifies the role of Safranal and Candesartan in prevention of the progression of Huntington's disease and its associated complications through their antioxidant, anti-inflammatory, anti-apoptotic and neuromodulator effects. I think the manuscript is of good idea. However, the manuscript needs some changes to be taken into consideration.

• The authors should clarify in the abstract the timing and duration of drugs administration as well as 3-NP. The authors also could add a figure for the experimental design in the methodology.

• Kindly modify the Guide for the Care and Use of Laboratory Animals 129 published by the US National Institute of Health (NIH Publication No. 85–23, revised 1996) to a more recent one (NIH Publication No. 85-23, revised 2011).

• The authors should explain the reason for using these large number of animals (40 rats per group)

• The authors should provide references for estimation of complexes I and II

• The authors could add a recent reference demonstrating the role of safranal in neurodegenerative disease (PMID: 33611726).

• Kindly check that all abbreviations are defined before being used like RIPA, TBS, ……etc

• The manuscript needs to be edited for few awkward sentences and grammar mistakes

**Reviewer #2:** REVIEW REPORT

Manuscript number: PONE-D-23-06048

Full Title: Effect of safranal or candesartan on 3-nitropropionicacid-induced biochemical, behavioral and histological alterations in a rat model of Huntington’s disease

Authors assessed the effect of safranal and candesartan on 3-nitropropionicacid-induced biochemical, behavioral and histological alterations in a rat model of Huntington’s disease. This is an interesting subject area. Based on the findings of the research, authors concluded that safranal and candesartan may prevent or delay the progression of HD and its associated impairments through their antioxidant, anti-inflammatory, anti-apoptotic and neuromodulator effects. Although the manuscript has been well written, some clarifications or inputs need to be made in order to greatly improve the quality of this manuscript. Below are some of the comments.

Abstract

1. A statement on the potential neuroprotective effects of safranal and candesartan should be inserted before the aim to justify why these drugs were chosen.

2. The doses of the test compounds should be well written. Are authors referring to 50 mg/kg for safranal and 1 mg/kg for candesartan? In line 48, authors should indicate the route of administration for 3-NP.

3. The sentence in Lines 48-50 is quite confusing. Description of methods should always be precise and easy to apprehend. Authors should kindly revise.

Introduction

4. In line 65, CAG should be defined. It is always important to define abbreviations in a text on first mention. The sentence captured in lines 88-90 is incomplete. Please revise.

Materials and Methods

5. In addition to weight, can authors state the age of animals. Were the animals housed individually or in groups during the acclimatization period? If in groups, how many were housed in a group? Also provide cage dimensions. Authors should indicate the start times of the light and dark cycle.

6. Number of animals (40) in a group are too many, bringing about issues regarding ethics. Why 40? How were drug solutions prepared and in what volume were they administered? What was the rationale of administering paraffin oil and CMC to some groups of rats? Why wasn’t a naïve control group introduced. How were drugs administered orally? What informed the choice of doses for safranal and candesartan? In addition, why did authors decide to use single doses for the test compounds? Why did authors choose to administer test compounds for 9 days?

7. Why did group 5 receive 3-NP from day 3 and not day 1? The statement “The sixth group (Safr+3-NP) received IP injection of 50 mg Safr/kg/day for 9 days followed by IP injection of 20 mg 3-NP/kg/day from the 3rd day to the 9th day’ is difficult to appreciate. Same can be said for group 7. Authors should clarify exactly how the dosing of 3-NP plus test compounds was done.

8. With regards to the behavioural tests performed, can authors indicate the exact start day and time of the experiments. For example, EPM was conducted on day 10 or 11 and at 9 a.m. after start of treatment etc.

9. OFT. Why was wood and not Plexiglas used for the OF apparatus? Authors indicate that the OFT was conducted in morning daylight. What was the exact time? Dimension for the OF apparatus: (90 x 90 x 25 cm3) or (90 cm x 90 cm x 90 cm) not (90 x 90 x 25 cm). same applies to the squares. What action was taken after the 5-minute session if an animal defecated in the arena? How were the parameters measured? The OFT is a conventional approach/avoidance paradigm in which anxiety and exploration are simultaneously evoked by a novel setting. An increase in activity or time spent in the center of the open field indicates reduction in anxiety and/or increase in exploration. It would have therefore been very important for authors to measure central activity.

10. EPM. The description indicates that anxiety behaviour was measured, not cognitive function. To assess cognitive function, the EPM transfer latency can be used. Please refer to literature. Was the apparatus made of wood or Plexiglas? How were the parameters recorded? With regards to dimensions, refer to comments on OF.

11. Experiments to measure movement behaviour should have been performed to assess the effects of safranal and Candesartan on motor abnormalities since this is an important trait of HD. Refer to Sidhu et al, 2018.

12. With regards to the one-way ANOVA, which post hoc test was used for multiple comparisons.

Results

13. What were the observed effects after 3-NP administration?

14. In line 277, which corresponding control group are authors referring to?

15. Statistical analysis e.g. p values are missing in the description.

16. Too many tables. Presentation of results as a blend of figures and tables would have been the best. Figures most often are easy to interpret. Ill suggest that some results from the behavioural tests are presented as figures.

17. For all tables, the statement “Data is displayed as mean ± standard error (SE)” can be captured in the legend.

18. For all tables, authors should note that the significant differences that exist with regards to the various comparisons featured in the legend have not been duly captured. For instance, e: represents significant difference as compared to 3-NP-treated group. Looking, at the values for PO or CMC in table 2, there should be significance in some of the parameters.

19. With regards to figs 1-4 (Photomicrograph of the striatum for the various treatments), authors can show in the micrographs, the normal and histological changes of neurons in the striatum for better appreciation.

Discussion

20. Authors made reference to a study by Fotoohi et al, regarding the beneficial effects of Safranal on movement defects induced by 3-NP during the two weeks of experiment. The doses used in this study are quite higher that that study (0.75, 1.5 and 3.0 mg/kg)? Why didn’t authors use the same or similar doses since beneficial effects were achieved with them.

21. There is silence on the relevance of assessing the effects of the test compounds in the EPM and OFT. A detailed discussion is needed. As stated earlier, EPM test measures anxiety behaviour but can also be used to measure cognitive function in the transfer latency test. This wasn’t done. It is therefore surprising authors are relating EPM to cognitive impairment.

22. In paragraph 2, authors conclude by stating “the current results showed that 3-NP injection significantly upregulated the mRNA levels of MAO-A and –B.” What about the effects of test compounds? Authors should focus their discussion on the benefits of test drugs based on the results.

6. PLOS authors have the option to publish the peer review history of their article (what does this mean?). If published, this will include your full peer review and any attached files.

Reviewer #1: No

Reviewer #2: No

---

## [Author Response · Author response to Decision Letter 0]

16 Jul 2023

Ref.: Ms. No. PONE-D-23-06048

Title: “Effect of safranal or candesartan on 3-nitropropionicacid-induced biochemical, behavioral and histological alterations in a rat model of Huntington’s disease”.

Journal: PLOS ONE

Dear editor-in chief, 

Respected Editor of The Journal of PLOS ONE

We appreciate the valuable inputs given by the respected reviewers which helped us greatly to improve the manuscript. We have tackled all the recommended changes. Below, you will find answers item-by-item to the reviewers’ comments:

Dear respected reviewers, 

On behalf of my coauthors, I am so grateful to you for your thorough revision and helpful insights that will certainly support our manuscript. 

Reviewer #1:

Comment #1: The authors should clarify in the abstract the timing and duration of drugs administration as well as 3-NP. The authors also could add a figure for the experimental design in the methodology.

Response: In the revised manuscript, under Abstract section, page 2; lines 49-53, all the timings and durations of drug and 3-NP administrations were illustrated.

In the revised manuscript, page 5, line 132, fig 1 was added as a diagrammatic representation of the experimental design.

Comment #2: Kindly modify the Guide for the Care and Use of Laboratory Animals 129 published by the US National Institute of Health (NIH Publication No. 85–23, revised 1996) to a more recent one (NIH Publication No. 85-23, revised 2011).

Response: In the revised manuscript, under M&M, page 5, line 138, the necessary changes were considered and the (NIH Publication No. 85–23, revised 1996) was replaced by the more recent one (NIH Publication No. 85-23, revised 2011).

Comment #3: The authors should explain the reason for using these large number of animals (40 rats per group).

Response: All the studied parameters were measured in the striatum that is a very small part of the brain weighing (50-100 mg). In order to measure each of the studied parameters striatum should be homogenized in a special buffer. For example, oxidative parameters (1.15% KCl), MAO (0.2M Phosphate buffer), ACHE (20mM Phosphate buffer), monoamines (n-acidified butanol),…etc, Accordingly, there was a need to take a large sample size to cover all the required parameters.

Comment #4: The authors should provide references for estimation of complexes I and II.

Response: In the revised manuscript, under M&M section, page 8, lines 217 and 230, the following references for the techniques of complexes I and II were cited.

Horowitz JD, Chong CR, Ngo DT, Sverdlov AL. Effects of acute hyperglycaemia on cardiovascular homeostasis: does a spoonful of sugar make the flow-mediated dilatation go down? J Thorac Dis 7: E607–E611, 2015. doi: 10.3978/j.issn.2072-1439.2015.12.40.

Ansari F, Yoval-Sánchez B, Niatsetskaya Z, Sosunov S, Stepanova A, Garcia C, Owusu-Ansah E, Ten V, Wittig I, Galkin A. Quantification of NADH:ubiquinone oxidoreductase (complex I) content in biological samples. J Biol Chem. 2021 Oct;297(4):101204. doi: 10.1016/j.jbc.2021.101204.

Comment #5: The authors could add a recent reference demonstrating the role of safranal in neurodegenerative disease (PMID: 33611726).

Response: In the revised manuscript, under Discussion section, line 479, page 23, the suggested reference (PMID: 33611726) is utilized to illustrate the role of safranal in neurodegenerative disease in comparison with the present study.

Comment #6: Kindly check that all abbreviations are defined before being used like RIPA, TBS, ……etc.

Response: in the revised manuscript, all the undefined abbreviations were defined and highlighted under M&M section, page 7 (lines 180, 189 & 192), page 8 (line 204-209) and page 9 (line 247 & 258), as well as under discussion section, page 23 (line 494), page 24 (lines 518 & 526) and page 26 (lines 578 & 586), as recommended.

Comment #7: The manuscript needs to be edited for few awkward sentences and grammar mistakes.

Response: All the manuscript items were carefully and thoroughly revised for language and grammatical mistakes, as per comment. 

Reviewer #2: 

Abstract

Comment #1: A statement on the potential neuroprotective effects of safranal and candesartan should be inserted before the aim to justify why these drugs were chosen.

Response: in the revised manuscript, page 2, lines 44-46, a statement was added as per request. "Safranal (Safr) that is found in saffron essential oil has antioxidant, anti-inflammatory and anti-apoptotic actions. Candesartan (Cands) is an angiotensin receptor blocker that has the potential to prevent cognitive deficits." 

Comment #2: The doses of the test compounds should be well written. Are authors referring to 50 mg/kg for safranal and 1 mg/kg for candesartan? In line 48, authors should indicate the route of administration for 3-NP.

Response: in the revised manuscript, page 2, lines51-53, the applied doses and routes of administration were clarified and highlighted.

Comment #3: The sentence in Lines 48-50 is quite confusing. Description of methods should always be precise and easy to apprehend. Authors should kindly revise.

Response: in the revised manuscript, page 2, lines 51-53, description of the experimental groups was revised as per comment.

Introduction

Comment #4: In line 65, CAG should be defined. It is always important to define abbreviations in a text on first mention. The sentence captured in lines 88-90 is incomplete. Please revise.

Response: in the revised manuscript, page 2, line 69, CAG was defined as cytosine–adenine–guanine. In addition, in the revised manuscript, page 3, lines 91-93, the sentence was revised and completed.

Materials and Methods

Comment #5: In addition to weight, can authors state the age of animals. Were the animals housed individually or in groups during the acclimatization period? If in groups, how many were housed in a group? Also provide cage dimensions. Authors should indicate the start times of the light and dark cycle.

Response: in the revised manuscript, page 4, line 113, the age of rats was stated. In page 4, lines 114-115, the number of animals per cage as well as the dimensions of the cage were stated. In page 4, lines 116-117, the starting time of light and dark cycle were stated.

Comment #6: Number of animals (40) in a group are too many, bringing about issues regarding ethics. Why 40? How were drug solutions prepared and in what volume were they administered? What was the rationale of administering paraffin oil and CMC to some groups of rats? Why wasn’t a naïve control group introduced. How were drugs administered orally? What informed the choice of doses for safranal and candesartan? In addition, why did authors decide to use single doses for the test compounds? Why did authors choose to administer test compounds for 9 days?

Response: 

1) Regarding te sample size, all the studied parameters were measured in the striatum that is a very small part of the brain weighing (50-100 mg). In order to measure each of the studied parameters striatum should be homogenized in a special buffer. For example, oxidative parameters (1.15% KCl), MAO (0.2M Phosphate buffer), ACHE (20mM Phosphate buffer), monoamines (n-acidified butanol),…etc, Accordingly, there was a need to take a large sample size to cover all the required parameters.

2) Regarding dose preparation of drugs:

a) Preparation of 3-NP (20 mg/kg)

Each 20mg of 3-NP was dissolved in 2.0 mL of 0.9% saline. The volume of administration was 2.0 mL/kg body weight.

b) Preparation of Safr (50mg/kg)

Each 50mg of Safr was dissolved in 2.0 mL of paraffin oil. The volume of administration was 2.0 mL/kg body weight.

c) Preparation of Cands (1mg/kg)

Each 1mg of Cands was dissolved in 2.0 mL of CMC. The volume of administration was 2.0 mL/kg body weight.

3) Regarding the used vehicles, note that CMC (carboxymethyl cellulose) is an inert vehicle that is suitable for giving homogenous suspension of Candsartan. CMC is safe for most animal species (EFSA et al., 2020). However, safranal is an oil that is miscible in paraffin oil (an inert vehicle). Parafin oil was safe to rodents (Shahat et al., 2022). Thus, there was no need to use naïve control group. 

4) Regarding the administration of signle doses, each test chemical was administered daily with the selected dose (a single dose per day).

5) In our study, the applied dose of 3-NP (20 mg/kg, IP), safranal (50 mg/kg, IP) and Candsartan (1mg/kg) were selected according to Sharma et al. (2012), Shahat et al. (2022), Ishrat et al., 2022, respectively. According to Sharma et al. (2018), an intraperitoneal injection of 20 mg of 3-NP/kg/day, for 7 days, is adequate to induce the HD model. Shahat et al. (2022) concluded that a daily injection of 50 mg safranal/kg, for 3 weeks, prevented the L-thyroxine-induced deleterious effects on the brain and improved its antioxidant status. According to Ishrat et al (2022), Cands could protect the rat's brain against stroke after 72h of administration.

6) In addition, Mehdizadeh et al (2013) reported that safranal extract at doses of 20 mg -160 mg/kg, for 9 days, exerted protective action against Isoproterenol-Induced Myocardial Infarction in Wistar Rats.

Shahat AS, Hassan WA, El-Sayed WM. N-Acetylcysteine and Safranal prevented the brain damage induced by hyperthyroidism in adult male rats. Nutr Neurosci. 2022;25(2):231-245.

EFSA Panel on Additives and Products or Substances used in Animal Feed (FEEDAP), Bampidis, V., Azimonti, G., Bastos, M. L., Christensen, H., Dusemund, B., Kos Durjava, M., Kouba, M., López-Alonso, M., López Puente, S., Marcon, F., Mayo, B., Pechová, A., Petkova, M., Ramos, F., Sanz, Y., Villa, R. E., Woutersen, R., Bories, G., Gropp, J., … Aquilina, G. (2020). Safety and efficacy of sodium carboxymethyl cellulose for all animal species. EFSA journal. European Food Safety Authority, 18(7), e06211. https://doi.org/10.2903/j.efsa.2020.6211

Mehdizadeh, R., Parizadeh, M. R., Khooei, A. R., Mehri, S., & Hosseinzadeh, H. (2013). Cardioprotective effect of saffron extract and safranal in isoproterenol-induced myocardial infarction in wistar rats. Iranian journal of basic medical sciences, 16(1), 56.‏

Sharma, M., Sharma, N., & Sharma, R. (2012). Neuroprotective effect of Zingiber officinale in 3-np-induced huntington disease. IOSR Journal of Pharmacy, 2(6), 61-70.‏

Ishrat T, Soliman S, Eldahshan W, Pillai B, Ergul A, Fagan SC. Silencing VEGF-B diminishes the neuroprotective effect of candesartan treatment after experimental focal cerebral ischemia. Neurochem Res. 2018;43, 1869–1878. 

Comment #7: Why did group 5 receive 3-NP from day 3 and not day 1? The statement “The sixth group (Safr+3-NP) received IP injection of 50 mg Safr/kg/day for 9 days followed by IP injection of 20 mg 3-NP/kg/day from the 3rd day to the 9th day’ is difficult to appreciate. Same can be said for group 7. Authors should clarify exactly how the dosing of 3-NP plus test compounds was done.

Response: The study aims to investigate the potential protective roles of Safr and Cands. Accordingly, the authors tend to start the treatments before the induction of HD model with two days. On the third day, a dose of 20 mg/kg for 7 days is enough to induce HD model in Wister rats, according to Sharma et al. (2012).

In the revised manuscript, page 4, lines 128 -131, the description of groups 6 and 7 was revised as per comment.

Sharma, M., Sharma, N., & Sharma, R. (2012). Neuroprotective effect of Zingiber officinale in 3-np-induced huntington disease. IOSR Journal of Pharmacy, 2(6), 61-70.‏

Comment #8: With regards to the behavioural tests performed, can authors indicate the exact start day and time of the experiments. For example, EPM was conducted on day 10 or 11 and at 9 a.m. after start of treatment etc.

Response: in the revised manuscript, page 5-6, lines 145, 155, 161 and 164, the starting time of each experiment was stated and highlighted.

Comment #9: OFT. Why was wood and not Plexiglas used for the OF apparatus? Authors indicate that the OFT was conducted in morning daylight. What was the exact time? Dimension for the OF apparatus: (90 x 90 x 25 cm3) or (90 cm x 90 cm x 90 cm) not (90 x 90 x 25 cm). same applies to the squares. What action was taken after the 5-minute session if an animal defecated in the arena? How were the parameters measured? The OFT is a conventional approach/avoidance paradigm in which anxiety and exploration are simultaneously evoked by a novel setting. An increase in activity or time spent in the center of the open field indicates reduction in anxiety and/or increase in exploration. It would have therefore been very important for authors to measure central activity.

Response: in the revised manuscript, page 5, line 143, the dimensions of OF apparatus were corrected. In page 5, lines 147-150, the action after each session was stated. Regarding the central activity, that is very interesting and should be taken into consideration but here in our study we focused on using OFT to measure the effects on general motor activity.

According to Okaichi et al., (2006), wooden OF apparatus is suitable for measuring ambulation frequency. Lamprea et al. (2008) also used wooden OF to measure locomotor activity.

Okaichi, Y., Amano, S., Ihara, N., Hayase, Y., Tazumi, T., & Okaichi, H. (2006). Open‐field behaviors and water‐maze learning in the F substrain of Ihara epileptic rats. Epilepsia, 47(1), 55-63.‏

Lamprea, M. R., Cardenas, F. P., Setem, J., & Morato, S. (2008). Thigmotactic responses in an open-field. Brazilian Journal of Medical and Biological Research, 41, 135-140.‏

Comment #10: EPM. The description indicates that anxiety behaviour was measured, not cognitive function. To assess cognitive function, the EPM transfer latency can be used. Please refer to literature. Was the apparatus made of wood or Plexiglas? How were the parameters recorded? With regards to dimensions, refer to comments on OF.

Response: in the revised manuscript, page 6, line 152, the aim of the EPM test was changed as to copy with measured parameters of EPM. As open arm time and number of entries are measures of anxiety while total or/and close arm entries represent measures of the spontaneous motor activity (Walf & Frye, 2007).

In the revised manuscript, page 5, lines 152-156, the apparatus material, dimensions and how parameters measured are all listed and highlighted as per comment. 

Walf, A. A., & Frye, C. A. (2007). The use of the elevated plus maze as an assay of anxiety-related behavior in rodents. Nature protocols, 2(2), 322–328. https://doi.org/10.1038/nprot.2007.44

Comment #11: Experiments to measure movement behaviour should have been performed to assess the effects of safranal and Candesartan on motor abnormalities since this is an important trait of HD. Refer to Sidhu et al, 2018.

Response: of course, measuring the movement behaviour is so important to assess the effectiveness of the used drugs against the HD model. Thus, in the present study, we utilized open field test, as a test for general locomotion (Osmon et al., 2018).

Osmon, K. J., Vyas, M., Woodley, E., Thompson, P., & Walia, J. S. (2018). Battery of Behavioral Tests Assessing General Locomotion, Muscular Strength, and Coordination in Mice. Journal of visualized experiments : JoVE, (131), 55491. https://doi.org/10.3791/55491

Comment #12: With regards to the one-way ANOVA, which post hoc test was used for multiple comparisons.

Response: in the revised manuscript, page 9, line 256, the applied post hoc test was illustrated as duncan's test.

Results

Comment #13: What were the observed effects after 3-NP administration?

Response: in the revised manuscript, RESULT section, pages 9, 12, 16 and 17, each paragraph started with describing the effect of 3-NP on the studied parameters. 

Comment #14: In line 277, which corresponding control group are authors referring to?

Response: in the revised manuscript, page 11, line 291, the corresponding control group here refers to the second group (control for Cands and 3-NP), as clarified in experimental design, page 4-5. 

Comment #15: Statistical analysis e.g. p values are missing in the description.

Response: in the revised manuscript, result section, action was taken, p-values were added to the comments.

Comment #16: Too many tables. Presentation of results as a blend of figures and tables would have been the best. Figures most often are easy to interpret. Ill suggest that some results from the behavioural tests are presented as figures.

Response: in the revised manuscript, pages 10 – 13, the tables of behavioral tests were replaced by figures 2-5.

Comment #17: For all tables, the statement “Data is displayed as mean ± standard error (SE)” can be captured in the legend.

Response: in the revised manuscript, action was taken in all tables.

Comment #18: For all tables, authors should note that the significant differences that exist with regards to the various comparisons featured in the legend have not been duly captured. For instance, e: represents significant difference as compared to 3-NP-treated group. Looking, at the values for PO or CMC in table 2, there should be significance in some of the parameters.

Response: statistical test was changed to Duncan's test for better clarification.

Comment #19: With regards to figs 1-4 (Photomicrograph of the striatum for the various treatments), authors can show in the micrographs, the normal and histological changes of neurons in the striatum for better appreciation.

Response: In the figures file, the arrows were added to the figures to illustrate the normal structure of the striatum as well as the histopathological changes in the experimental groups.

Discussion

Comment #20: Authors made reference to a study by Fotoohi et al, regarding the beneficial effects of Safranal on movement defects induced by 3-NP during the two weeks of experiment. The doses used in this study are quite higher that that study (0.75, 1.5 and 3.0 mg/kg)? Why didn’t authors use the same or similar doses since beneficial effects were achieved with them.

Response: In our study, the applied dose of 3-NP (20 mg/kg, IP) and safranal (50 mg/kg, IP) and were selected according to Sharma et al. (2012) and Shahat et al. (2022), respectively. According to Sidhu et al. (2018), an intraperitoneal injection of 20 mg of 3-NP/kg/day, for 7 days, is adequate to induce the HD model. Shahat et al. (2022) concluded that a daily injection of 50 mg safranal/kg, for 3 weeks, prevented the L-thyroxine-induced deleterious effects on the brain and improved its antioxidant status.

Shahat AS, Hassan WA, El-Sayed WM. N-Acetylcysteine and Safranal prevented the brain damage induced by hyperthyroidism in adult male rats. Nutr Neurosci. 2022;25(2):231-245.

Sharma, M., Sharma, N., & Sharma, R. (2012). Neuroprotective effect of Zingiber officinale in 3-np-induced huntington disease. IOSR Journal of Pharmacy, 2(6), 61-70.‏

Comment #21: There is silence on the relevance of assessing the effects of the test compounds in the EPM and OFT. A detailed discussion is needed. As stated earlier, EPM test measures anxiety behaviour but can also be used to measure cognitive function in the transfer latency test. This wasn’t done. It is therefore surprising authors are relating EPM to cognitive impairment.

Response: in the revised manuscript, page 22, lines 497-499, the estimated output of EPM was changed as per advice.

Comment #22: In paragraph 2, authors conclude by stating “the current results showed that 3-NP injection significantly upregulated the mRNA levels of MAO-A and –B.” What about the effects of test compounds? Authors should focus their discussion on the benefits of test drugs based on the results.

Response: in the revised manuscript, page 25, lines 540-543 as well as page 26, lines 572-574, the discussion of safr and Cands was revised and highlighted.

We hope that we have answered all the comments raised by the respected referees and followed their recommendations and adopted the manuscript on journal's guidelines/style. We think that our manuscript has been considerably improved because of these thoroughly revisions. We hope that our revised manuscript “Effect of safranal or candesartan on 3-nitropropionicacid-induced biochemical, behavioral and histological alterations in a rat model of Huntington’s disease” is now suitable for publication in your respected journal “PLOS ONE”.

We should like to take this opportunity to thank the editor for his dedication and support. We look forward to hearing from you regarding our submission.

Thank you for understanding and support

Respectfully

Sherine Maher Rizk, PhD 

Biochemistry Department

Faculty of Pharmacy

Cairo University 

E-mail: nagwa.mahmoud@pharma.cu.edu.eg

---

## [Decision Letter · Decision Letter 1]

23 Aug 2023

PONE-D-23-06048R1Effect of safranal or candesartan on 3-nitropropionicacid-induced biochemical, behavioral and histological alterations in a rat model of Huntington’s diseasePLOS ONE

Dear Dr. Risk,

Thank you for submitting your manuscript to PLOS ONE. After careful consideration, we feel that it has merit but does not fully meet PLOS ONE’s publication criteria as it currently stands. Therefore, we invite you to submit a revised version of the manuscript that addresses the points raised during the review process.

We look forward to receiving your revised manuscript.

Kind regards,

Yasmina Abd‐Elhakim

Academic Editor

PLOS ONE

Journal Requirements:

Reviewers' comments:

Reviewer's Responses to Questions

**Comments to the Author**

1. If the authors have adequately addressed your comments raised in a previous round of review and you feel that this manuscript is now acceptable for publication, you may indicate that here to bypass the “Comments to the Author” section, enter your conflict of interest statement in the “Confidential to Editor” section, and submit your "Accept" recommendation.

Reviewer #1: All comments have been addressed

Reviewer #2: (No Response)

2. Is the manuscript technically sound, and do the data support the conclusions?

Reviewer #1: Yes

Reviewer #2: Yes

3. Has the statistical analysis been performed appropriately and rigorously? 

Reviewer #1: Yes

Reviewer #2: Yes

4. Have the authors made all data underlying the findings in their manuscript fully available?

Reviewer #1: Yes

Reviewer #2: Yes

5. Is the manuscript presented in an intelligible fashion and written in standard English?

Reviewer #1: Yes

Reviewer #2: Yes

6. Review Comments to the Author

Reviewer #1: (No Response)

Reviewer #2: Introduction

1. The sentence captured in lines 91-93 ‘’Ang II type 1 receptor (AT1R) blockers (ARBs) have been reported to be beneficial in attenuating cognitive deficits observed in Alzheimer’s disease, Parkinson’s disease, vascular’’ is incomplete. Please revise.

Materials and Methods

2. Details regarding preparation of the various drug solutions should be outlined in the manuscript (This is captured in your response)

3. For the experimental design/procedure, authors can represent the various treatments as Safr 50 mg/kg/day rather than 50 mg Safr/kg/day.

4. For the various dimensions (cm3), 3 should be in superscript.

5. With regards to the behavioural testing, authors indicate that the training session for the passive avoidance test started at 6:00 pm on training day 9 in which a 0.5 mA with 50 Hz foot electric shock was delivered for two seconds. The behavioural assessment in the EPM & OFT was however done on day 10 of treatment. Couldn’t this have influenced the results seen with regards to the EPM and OFT? In line 149, can authors state the % alcohol used?

6. In Fig 1, the experimental design should capture a detailed sequential summary of all experiments from start to finish. This includes treatments, behavioural assessments & sample analysis.

Results

7. What were the observed effects after 3-NP administration?

8. Capital letters are used to represent the outcomes of the statistical analysis. Can authors use symbols instead? Maybe *, #, etc.

9. Symbol for seconds is “s”.

10. The y-axis of the various figures can be numbered without decimals.

11. Authors used different in-text citations for the figures. Eg. fig 2, fig(5), or Fig. 5. Authors should refer to the journal’s guidelines and be consistent.

Discussion

12. There is silence on the relevance of assessing the effects of the test compounds in the EPM and OFT.

7. PLOS authors have the option to publish the peer review history of their article (what does this mean?). If published, this will include your full peer review and any attached files.

Reviewer #1: No

Reviewer #2: No

---

## [Author Response · Author response to Decision Letter 1]

16 Oct 2023

Ref.: Ms. No. PONE-D-23-06048R1

Title: “Effect of safranal or candesartan on 3-nitropropionicacid-induced biochemical, behavioral and histological alterations in a rat model of Huntington’s disease”.

Journal: PLOS ONE

Dear editor-in chief, 

Respected Editor of The Journal of PLOS ONE

We appreciate the valuable inputs given by the respected reviewers which helped us greatly to improve the manuscript. We have tackled all the recommended changes. Below, you will find answers item-by-item to the reviewers’ comments:

Dear respected reviewers, 

On behalf of my coauthors, I am so grateful to you for your thorough revision and helpful insights that will certainly support our manuscript. 

Reviewer #2:

Introduction

Comment #1: The sentence captured in lines 91-93 ‘’Ang II type 1 receptor (AT1R) blockers (ARBs) have been reported to be beneficial in attenuating cognitive deficits observed in Alzheimer’s disease, Parkinson’s disease, vascular’’ is incomplete. Please revise.

Response: In the revised manuscript, Introduction section, page 3, lines 92-94, the captured sentence was revised as follows:

"Ang II type 1 receptor (AT1R) blockers (ARBs) have been reported to be useful in diminishing cognitive deficits linked to Post-Stroke Cognitive Impairment (PSCI), Alzheimer’s Disease, Parkinson’s Disease, and Vascular Cognitive Impairment (VCI)"

Materials and Methods

Comment #2: Details regarding preparation of the various drug solutions should be outlined in the manuscript (This is captured in your response).

Response: In the revised manuscript, M&M section, page 4, lines 114-121, the details of drug preparations were stated for each of the applied drugs.

Comment #3: For the experimental design/procedure, authors can represent the various treatments as Safr 50 mg/kg/day rather than 50 mg Safr/kg/day.

Response: In the revised manuscript, M&M section, page 5, lines 134-140, the rcommended comments in the experimental design/procedure were considered.

Comment #4: For the various dimensions (cm3), 3 should be in superscript.

Response: In the revised manuscript, M&M section, page 4 (line 125), page 6 (lines 154- 171), action was taken as per comment.

Comment #5: With regards to the behavioural testing, authors indicate that the training session for the passive avoidance test started at 6:00 pm on training day 9 in which a 0.5 mA with 50 Hz foot electric shock was delivered for two seconds. The behavioural assessment in the EPM & OFT was however done on day 10 of treatment. Couldn’t this have influenced the results seen with regards to the EPM and OFT? In line 149, can authors state the % alcohol used?

Response: in the revised manuscript, M&M section (fig 1), pages 5, line 139, in the summary of the experimental design, a flowchart was added to show that each of the behavioural tests (OFT, EPM & PA) was conducted using different set of the experimental animals (n=7, per group, per test). 

Actually, we had two options; option #1 is to conduct the OFT and EPM on the 10th day, then to run the training session of PA, on the 11th day, then the main experiment of PA on the 12th day. Accordingly, three days would be passed after the last treatment. thus, the results of PA may not be representative for the conducted treatment. 

Option #2 is to conduct the behavioural experiments using different set of the experimental animals (n=7, per group, per test), to avoid the effect of the electrical shock in the PA on the results of the other behavioral tests. We chosen the second option, because it gave us the opportunity to apply all the beahvioural tests on the 10th day and to avoid that any of the studied behaviors may be affected by the other behavioural tests.

In the revised manuscript, M&M section, page 6, line 161, the % of alcohol was stated as 70%.

Comment #6: In Fig 1, the experimental design should capture a detailed sequential summary of all experiments from start to finish. This includes treatments, behavioural assessments & sample analysis.

Response: in the revised manuscript, M&M section, page 5, lines 139-140, the detailed sequential summary of all the experiments was stated as per comment.

Results

Comment #7: What were the observed effects after 3-NP administration?

Response: in the revised manuscript, Results section, page 10, lines 277-280, a brief description of the the effect of 3-NP injection was written. In addition, in the subsequent paragarphs, the detailed description of the effect of 3-NP inection was written and highlighted in yellow.

Comment #8: Capital letters are used to represent the outcomes of the statistical analysis. Can authors use symbols instead? Maybe *, #, etc.

Response: As clarified in page 10, line 267, under M&M-statistical analysis, the applied post-hoc test was Duncan's test to study the statistical homogeneity among all the experimental groups. Accordingly, letters are best fitting to represent the outcomes. However, to use symbols like "*, #,…", are best fitting for the outcome of other statistical post-hoc tests like for example tukey HSD and LSD. Duncan's test gives us the opportunity to compare among all groups, using Alphabet letters (in ascending or desecending order), especially that the groups have multiple controls. However, the outcomes of LSD and HSD may cause some confusion because there will be at least 6 symbols (as compared to the PO-group, the CMC-group, the Safr-alone group, the Cands-alone group, 3NP-group and Safr+3NP group) 

Comment #9: Symbol for seconds is “s”.

Response: In the revised manuscript, M&M section, page 6-7 (lines 157, 174-176), as well as in Results section, page 12 (Fig 3), page 14 (Fig 4) and page 15 (Fig 5), action was taken as per comment.

Comment #10: The y-axis of the various figures can be numbered without decimals.

Response: In the revised manuscript, Results section, page 11 (Fig 2), page 12 (Fig 3), page 14 (Fig 4) and page 15 (Fig 5), all decimals were removed as per comment.

Comment #11: Authors used different in-text citations for the figures. Eg. fig 2, fig(5), or Fig. 5. Authors should refer to the journal’s guidelines and be consistent.

Response: In the revised manuscript, pages 4-5 (Fig 1), under M&M section as well as in page 10-15 (Figs 2-5), under the results section, all Figs were adjusted according to the journal guidelines.

Discussion

Comment #12: There is silence on the relevance of assessing the effects of the test compounds in the EPM and OFT.

Response: In the revised manuscript, in Discussion section, page 26 (lines 565-576) and page 27 (lines 590-602), action was taken as per comment.

We hope that we have answered all the comments raised by the respected referees and followed their recommendations and adopted the manuscript on journal's guidelines/style. We think that our manuscript has been considerably improved because of these thoroughly revisions. We hope that our revised manuscript “Effect of safranal or candesartan on 3-nitropropionicacid-induced biochemical, behavioral and histological alterations in a rat model of Huntington’s disease” is now suitable for publication in your respected journal “PLOS ONE”.

We should like to take this opportunity to thank the editor for his dedication and support. We look forward to hearing from you regarding our submission.

Thank you for understanding and support

Respectfully

Sherine Maher Rizk, PhD 

Biochemistry Department

Faculty of Pharmacy

Cairo University 

E-mail: sherine.abdelaziz@cu.edu.eg

---

## [Editor Report · Decision Letter 2]

18 Oct 2023

Effect of safranal or candesartan on 3-nitropropionicacid-induced biochemical, behavioral and histological alterations in a rat model of Huntington’s disease

PONE-D-23-06048R2

Dear Dr. Risk,

We’re pleased to inform you that your manuscript has been judged scientifically suitable for publication and will be formally accepted for publication once it meets all outstanding technical requirements.

Kind regards,

Yasmina Abd‐Elhakim

Academic Editor

PLOS ONE
---

## [Editor Report · Acceptance letter]

24 Oct 2023

PONE-D-23-06048R2 

Effect of safranal or candesartan on 3-nitropropionicacid-induced biochemical, behavioral and histological alterations in a rat model of Huntington’s disease 

Dear Dr. Rizk:

I'm pleased to inform you that your manuscript has been deemed suitable for publication in PLOS ONE. Congratulations! Your manuscript is now with our production department. 

Kind regards, 

on behalf of

Prof. Dr. Yasmina Abd‐Elhakim 

Academic Editor

PLOS ONE